# Technical note: Flow cytometry assays for the detection, counting, and cell-sorting of polyphosphate-accumulating bacteria

Clémentin Bouquet[1], Hermine Billard[1,2], Cécile C. Bidaud[3], Jonathan Colombet[1,2], Young-Tae Chang[4], Joan Artigas[1], Isabelle Batisson[1,] Karim Benzerara[3], Fériel Skouri-Panet[3], Elodie Duprat[3], Anne-Catherine Lehours[1]

[1]Université Clermont Auvergne, CNRS, LMGE, F-63000 Clermont-Ferrand, France
[2]UCA Partner, Cytometry, Sort and Transmission Electronic Microscopy (CYSTEM) platform, F-63000 Clermont-Ferrand, France
[3]Sorbonne Université, Muséum National d'Histoire Naturelle, UMR CNRS 7590 – Institut de Minéralogie, de Physique des Matériaux et de Cosmochimie (IMPMC), Paris, France
[4]Department of Chemistry, Pohang University of Science and Technology (POSTECH), Pohang 37673, Republic of Korea

Correspondence to: Anne-Catherine Lehours (a-catherine.lehours@uca.fr)

**Abstract.** In the context of the ecological sustainability of phosphorus, emerging evidence for the ubiquitous presence of polyphosphate-accumulating bacteria in natural environments invites efforts to reveal their roles in the biogeochemical cycle of phosphorus. This requires high-throughput methods to characterise their structure and dynamics in ecosystems. A promising strategy is to combine the staining of intracellular polyphosphate granules and their subsequent detection by flow cytometry, enabling rapid data acquisition. In this study, we evaluated the potential of this approach by testing various factors that could affect the efficiency and specificity of polyphosphate labelling. Most of our experiments were performed using the 4'6-diamidino-2-phenylindole dye (DAPI). However, we also carried out a preliminary study using the synthetic fluorochrome JC-D7, a new selective fluorescent dye used for the specific labelling of endogenous polyphosphate in living cells. The assays were performed on *Tetrasphaera elongata,* a Gram-positive bacterium, known to accumulate large amounts of intracellular polyphosphates. We also used six bacterial strains belonging to different phyla, in particular a Gram-negative bacterial strain belonging to the genus *Pseudomonas*, which is characterised by low levels of cellular polyphosphate. The potential of flow cytometry to quantify and sort polyphosphate-accumulating bacteria in complex environmental samples, including soil, freshwater and sediments, was also examined. Our tests provide useful information for the design of future experiments and highlight the potential pitfalls and limitations of detecting polyphosphate-accumulating bacteria using the cytometric approach. We also show that JC-D7 is a promising dye for achieving these objectives, particularly for enumerating polyphosphate-accumulating bacteria from environmental samples.

## 1 Introduction

Since the 'green revolution' of the 1960s, the phosphorus (P) contained in geological deposits has been extracted in large quantities for the production of fertilisers, increasing the input of P into the biosphere fourfold compared to the pre-industrial era (Falkowski *et al.,* 2000). Over the same period, P storage in terrestrial and freshwater ecosystems increased dramatically (> 75 %; Bennett *et al.,* 2001). This P excess has led to deterioration in ecosystem services, notably the formation of hundreds of coastal dead zones associated with eutrophication (Diaz and Rosenberg, 2008). Paradoxically, and by analogy with 'peak oil', a 'phosphorus peak' is predicted by 2035

(Cordell *et al.,* 2009; 2011). In order to increase the sustainability of P-resources managements, it is crucial to significantly improve our knowledge about the detailed processes, fluxes and reservoirs involved in the biogeochemical cycle of P. Microorganisms have been shown to be major actors in modern and past cycles of P either as reservoirs and/or catalysts of processes exchanging P between different reservoirs (Diaz *et al*., 2008). In this vein, there is emerging evidence of the unexpected and ubiquitous presence of polyphosphate-accumulating bacteria (PAB) in natural environments such as rivers, lakes, and soils, inviting efforts to reveal their unknown functions and roles in the context of P availability and cycling (Diaz *et al.,* 2008, Rivas-Lamelo *et al.,* 2017; Akbari *et al.,* 2021; Bidaud *et al*., 2022).

Intracellular polyphosphates (polyP) are ubiquitous biopolymers containing between a few and hundreds of orthophosphate residues linked together by phosphoanhydride bonds. Monovalent or divalent metal elements, such as $Mg^{2+}$, $K^+$, $Ca^{2+}$ and $Na^+$ can act as counterions in polyP polymers, forming complexes with negatively charged phosphate residues (Akbari *et al.,* 2021). PolyP are found in representatives of all kingdoms of living organisms and every cell type in nature (Lorenzo-Orts *et al.,* 2020, Akbari *et al.,* 2021). Likely a key agent in evolution from prebiotic time (Brown and Kornberg, 2004; Lorentzo-Orts *et al.,* 2020), the functions of polyP in cells of contemporary organisms are many and varied (Konberg *et al.,* 1999). PolyP can serve as a source of energy; as a phosphorylating agent for alcohols, including sugars, nucleosides, and proteins; and as a means of activating the precursors of fatty acids, phospholipids, polypeptides, and nucleic acids (Rao *et al.,* 2009). In PAB, polyP storage is exacerbated and polyP granules, which are spherical aggregates, can account for up to 20 % of the dry weight of these bacteria. PAB cells accumulate these polymers at cellular concentrations up to millimolar, for example, as an energy reserve to adapt and survive environmental gradients or to scavenge nutrients (Martin *et al.,* 2014). PolyP accumulation in PAB can have an impact on P biogeochemistry, and PAB are expected to play critical roles as reservoirs or catalysts for P exchange between the geosphere and the biosphere (Diaz *et al*., 2008; Cosmidis *et al*., 2014); yet they are still missing from the global P cycle models.

Numerous methodologies to quantify and characterise polyP have been developed, including chemical, biological, molecular, and microscopic approaches (Majed *et al.,* 2012). Most conventional analytical methods (e.g. electron ionisation mass spectrometry) require extensive sample preparation, pre-treatment, and pre-fractionation procedures. Advanced analytical techniques, such as nuclear magnetic resonance, Raman, Raman-FISH (Fernando *et al*., 2019), and X-ray spectromicroscopy, require much less pre-treatment and allow polyP to be characterised with high molecular and spatial resolution (<µm). Although the potential of these approaches in environmental and biological research is clear, their use remains limited due to the cost and accessibility of analysis instruments. Photometric approaches offer an interesting alternative to the methods discussed above and, the most relevant to date, are based on the interaction between polyP and the 4'6-diamidino-2-phenylindole fluorochrome (DAPI) (Martin and Van Mooy, 2013). The binding of DAPI to polyP shifts the wavelength of maximum emission from DAPI and, as a result, the intensity of fluorescence at this shifted wavelength is proportional to the intracellular polyP concentration (Tijssen *et al*., 1982, Aschar-Sobbi *et al*., 2008). This principle has played a decisive role in the visual identification of polyP granules in cells. These approaches which combine microscopic observations with DAPI labelling are simple but time-consuming techniques. Unveiling the environmental importance of PAB and their impact on the biogeochemical P cycle requires high-throughput methods to characterise their structure, dynamics, and function in complex and heterogeneous environmental samples at high spatial and temporal resolution (Günther *et al*., 2009). To this end, a promising strategy is to combine the specific staining of

intracellular polyP granules in PAB and their subsequent detection by flow cytometry (e.g. Zilles *et al.*, 2002; Günther *et al*., 2009; Terashima *et al.,* 2020).

Flow cytometry (FCM) is an essential tool in the field of environmental microbiology, enabling rapid data acquisition and multiparametric analyses. In combination with various dyes, FCM can be used to study communities and analyse thousands of microbial cells per second. The ability of fluorescence-activated cell sorting (FACS) also makes FCM a powerful technique for identifying and isolating microbial cells with particular characteristics (Zilles *et al.*, 2002; Terashima *et al.,* 2020). Although FCM has already been applied to detect polyP fluorescence induced by different dyes (Zilles *et al.*, 2002; Terashima *et al.,* 2020), there is still a lack of knowledge about the optimal parameters and the potential pitfalls of the FCM approach for the detection and counting of PAB in the environment.

To evaluate the detection and enumeration of PAB by FCM, we present here a detailed evaluation of a wide range of factors that may affect the quality of the fluorescence signal and, therefore, the efficiency of enumeration from DAPI staining of polyP. We also compare the staining of polyp with DAPI with that obtained with JC-D7, which is a benzimidazolinium dye. This novel polyP sensor has been shown to be suitable for staining polyP in living eukaryotic cells and tissues (Angelova *et al.,* 2014), and has recently been used to target polyP in yeast extracts (Deidert *et al.,* 2024) and planktonic environmental samples (Yang *et al*., 2024). The assays were performed using *Tetrasphaera elongata,* which can account for up to 30 % of the total bacteria in an enhanced biological phosphorus removal process (Nguyen *et al*., 2011). Our analyses also included six bacterial strains affiliated to different phyla, in particular a Gram-negative bacterium belonging to the genus *Pseudomonas*, which is characterised by low levels of cellular polyP. Furthermore, we carried out tests on microbial cells from soil, water, and lake sediment samples. Our work provides useful information for the design of future experiments by highlighting the potential applications, but also the pitfalls and limitations of PAB detection by FCM. It also highlights the JC-D7 dye as a promising fluorescent probe for PAB enumeration in environmental samples.

## 2    Material and methods

### 2.1.  Strains and culture conditions

*Tetrasphaera elongata* Lp2 (DSM 14184), a Gram-positive bacterium well known to accumulate large amounts of intracellular polyP (up to 30-35 % of the total biovolume of bacteria; Nguyen *et al.,* 2011), was used as a 'high polyP accumulation' control. Prior to this study, we screened our bacterial strain library at the Laboratoire Microorganismes: Génome et Environnement (LMGE) and identified the Gram-negative strain RX (99 % identity with *Pseudomonas trivialis*) as having a very low amount of intracellular polyP (i.e. only a few RX cells have polyP, and the polyP in these cells represents only a small fraction of the cell volume). The RX strain was used as a 'low polyP accumulation' control. We also compared polyP labelling with DAPI or JC-D7 dye using four Gram-negative strains affiliated with *Acinetobacter lwoffii* (*Pseudomonadales*). *Flavobacterium sp*. (*Bacteroidota*), *Pseudomonas sp.* (*Pseudomonadales*), *Stenotrophomonas rhizophila* (*Pseudomonadales*) and the Gram-positive strain *Microbacterium hydrocarbonoxydans* DSM 16089 (*Actinomycetota*). RX strain was isolated from a freshwater sample; the other Gram-negative strains were isolated from samples of decomposing leaf litter. All strains are preserved at the LMGE.

*T. elongata* cells were grown in NM-1 medium (pH 7.1) containing (per litre): glucose (0.5 g); peptone (0.5 g); monosodium glutamate (0.5 g); yeast extract (0.5 g); $K_2HPO_4$ (0.44 g); $(NH_4)_2SO_4$ (0.1 g); $MgSO_4$ x $7H_2O$ (0.1 g). The other strain cells, were grown in PCA medium (pH 7) containing (per litre): tryptone (5.0 g); yeast extract (2.5 g) and glucose (1.0 g). The culture media were autoclaved (20 min, 121°C) and then filtered through Stericup® vacuum filtration systems with a porosity of 0.2 μm. Cultures (10 % vol/vol inoculum) were incubated at 28 ° C in Falcon® aerobic cell culture flasks fitted with 0.2 μm hydrophobic membrane, in the dark, and shaken at 100 rpm. The kinetics of strain growth were monitored by measuring the optical density at 600 nm and subsequent analyses were performed during the exponential phase of growth.

### 2.2. Environmental samples

The water samples were collected in the water column of Lake Pavin at 54 m depth with an 8-liter horizontal Van Dorn bottle, and filtered by tangential flow filtration (0.2-μm cartridge) to yield concentrate. Sediments from the littoral zone of Lake Pavin (Auvergne, France) were sampled using a UWITEC corer (Mondsee, Austria) fitted with a polyvinyl chloride tube (1 m). The upper part (0-5 cm) of the sediment core was collected with a sterile 50 mL pipette. Samples of calcareous or colluvial soils, with neutral to alkaline pH, were taken from six agricultural parcels in Puy de Dôme (Auvergne, France) at a depth of 0-20 cm. Three of these parcels were subjected to conventional farming and the other three to biological farming. The relative bioavailability of inorganic orthophosphate ($PO_4$-P) in soil samples was estimated using the Olsen method (Olsen, 1982), which is widely used to determine bioavailable phosphate in soils with neutral to alkaline pH (Amini *et al.,* 2022).

To separate the microbial cells from the sediment or soil particles, 10 ml of 0.01 M sodium pyrophosphate buffer (pH 7.2) was added to 1 g of soil or sediment sample in a 15 mL Falcon® tube and the mixture shaken (280 rpm) at 4 °C for 30 minutes. The samples were placed for 1 min at 60 W in a sonication bath (Elmasonic S, Elma) and then centrifuged (2 min, 1500 g, 4 °C) (Lavergne *et al.,* 2014). The supernatant was collected and stored for 4 h at 4 °C until analysis.

### 2.3. Properties of fluorescent dyes and preparation of staining solutions

4'6-diamidino-2-phenylindole (DAPI), used at a final concentration of less than 1 μg mL$^{-1}$, is a fluorescent dye that strongly binds to DNA and the DAPI-DNA complex fluoresces blue, with a maximum emission at 460 nm, after excitation by an ultraviolet (UV, 350 nm) or violet (405 nm) laser (Button and Robertson, 2001). DAPI also forms complexes with polyP when used at high concentrations (3-50 μg mL$^{-1}$; Kulakova *et al.,* 2011). DAPI-polyP complexes emit yellow-green fluorescence (525-605 nm range) when excited by a violet laser (Allan and Miller, 1980) (Fig. S1). In the present study, a stock solution of DAPI (1 mg mL$^{-1}$, i.e. 2.85 mM) was prepared according to the manufacturer's instructions (Thermo Fischer Scientific, Rockford, USA). Solid DAPI (powder) was dissolved in ultrapure water, aliquoted, and stored at -20°C in the dark. For analysis, DAPI was diluted in the chosen buffer (HEPES, Tris-EDTA, PBS, or, McIlvaine; see section 2.4) before use.

The synthetic fluorochrome JC-D7 is identified as a polyP-specific marker (Angelova *et al.,* 2014). The JC-D7 dye excited at 405 nm shows blue-green fluorescence emission between 480 and 510 nm (Fig. S1). In this study, stock solutions (10 mM) of JC-D7 (Chemical Cellomics Laboratory, South Korea) were prepared in dimethyl sulfoxide (molecular biology grade DMSO, Merck, Darmstadt, Germany), aliquoted and stored at -20 °C in the dark. JC-D7 was diluted at 10 μM in HEPES buffer (see section 2.4) before use.

SYTO®62 fluorophore is a polymethine cyanine dye (cell permeable), that binds to nucleic acids. The DNA-SYTO®62 complex emits red fluorescence (676 nm) without spectral interaction with the polyP-DAPI or polyP-JC-D7 complexes, allowing the colocalization of DNA in PAB cells with polyP labelled with DAPI or JC-D7 (Fig. S1). In the present study, SYTO®62 stock solution (5 mM; Thermo Fischer Scientific, Rockford, USA) was stored at -20°C in the dark. For analysis, SYTO®62 was diluted at 1 µM in the chosen buffer (HEPES, Tris-EDTA, PBS, or McIlvaine; see section 2.4) before use.

### 2.4. Preparation of isotonic staining buffers

The isotonic buffers used were as follows:

- Phosphate buffer saline (PBS; 1X; pH 7.2) containing per litre: NaCl (8 g); KCl (0.2 g); $Na_2HPO_4$ (1.44 g), $KH_2PO_4$ (0.24 g) and MilliQ® water.

- 4-(2-hydroxyethyl)-1-piperazine ethane sulfonic acid buffer (HEPES; 20mM, pH 7.4) containing per litre: 0.48 g of HEPES (Sigma Aldrich, CAS: 7365-45-9) and MilliQ® water.

- Tris-hydrochloride and ethylenediaminetetraacetic acid buffer (Tris-EDTA; pH 7.4) containing a 10 mM Tris-HCl solution and a 1 mM EDTA solution (Merck KGaA, Darmstadt, Germany).

- Citrate phosphate buffer (McIlvaine; pH 7.2) containing per litre: 869.5 mL of a 0.2 M $Na_2HPO_4$ solution; 115.5 mL of a 0.1 M citric acid solution and MilliQ® water.

The buffers were sterilised by filtration on 0.2 µm (Minisart® syringe filter, Sartorius).

### 2.5. Treatments tested to define the optimum conditions for polyP staining with DAPI and sample storage conditions

To determine optimal conditions for intracellular polyP staining, *T. elongata* (TE) and RX cell culture samples were subjected to different treatments, including different types of staining buffer (PBS, HEPES, Tris-EDTA, McIlvaine), percentages of fixative used (2 % and 4 % of formaldehyde Merck KGaA, Darmstadt, Germany), storage temperatures (4°C, -20°C, -80°C), storage times (1 h and 2, 7, 14 days), and detergent addition (0 and 0.3 % Triton X100, Sigma CAS: 9002-93-1). Formaldehyde is a 37 % commercial solution (CAS: 50-00-0) then diluted directly in the sample.

### 2.6. Flow cytometric (FCM) analysis of PAB after DAPI staining

The samples (final volume 200 µL) were analysed using a BD LSR Fortessa™ X-20™ flow cytometer (BD BioSciences, San Jose, CA USA) in a three-laser configuration (405 nm, 50 mW; 488 nm, 60 mW; and 640 nm, 40 mW). Samples were diluted so that the event rate was less than 3,000 cell $s^{-1}$. Fluorescence intensity, total cell number, forward scatter (FSC) and side scatter (SSC) were recorded. The fluorescence from the DAPI-polyP complexes (excitation at 405 nm) and SYTO®62-DNA complexes (excitation at 640 nm) was collected with 530/30 nm and 670/14nm bandpass filters, respectively. Data were acquired and analysed on logarithmic scales using FACSDiva™ version 9.0 (BD Biosciences).

### 2.7. Fluorescence-activated cell sorting (FACS) enrichment of polyP-containing cells stained with DAPI

We analysed a water sample collected at a depth of 54 m in Lake Pavin and a culture sample of a fresh mixture of *T. elongata* and RX (50:50 abundance). The samples were centrifuged (4000 g, 20 min, 4 °C), resuspended in PBS solution, stained in the dark with DAPI (28.5 µM, 30 min) and SYTO®62 (1 µM, 10 min) and immediately

processed. Analysis and cell sorting were performed with a BD FACSAria™ Fusion SORP cell sorter equipped with a 70 µm nozzle and a 1.5 neutral density filter (BD BioSciences, San Jose, CA USA) in a three lasers configuration (405 nm, 50 mW; 488 nm, 50 mW; and 640 nm, 100 mW). A forward scatter (FSC) threshold of 200 was used, and DNA was monitored using a 640 nm excitation laser and 670/30 nm emission filter. PolyP fluorescence was monitored using a 405 nm violet laser and a 525/50 nm emission filter. Cell sorting was performed in purity mode and cells were sorted at a rate of approximately 1,500 cells s$^{-1}$ into two fractions: polyP+ (i.e. positive green fluorescence signal regarding the fluorescence intensity limit defined by the controls) and polyP- (i.e. absence of green fluorescence signal regarding the fluorescence intensity limit defined by the controls). Data were acquired and analysed on logarithmic scales using FACSDiva™ version 9.0 (BD Biosciences). Cell sorting was performed on a different machine from cell counting because the BD FACSAria™ Fusion SORP cell sorter, which is an extremely efficient cytometer for cell sorting, is cumbersome to setup. Cell counting was therefore performed on a BD LSR Fortessa™ X-20™, which is designed for this purpose. Both instruments have the same lasers and filters, making the analysis comparable, and internal quality control using fluorescent microbeads was used.

### 2.8. FCM and FACS controls

The settings for morphological (FSC and SSC) and fluorescence (DAPI, SYTO®62, JC-D7) parameters were set on the basis of samples not stained or independently stained by the different fluorochromes. Briefly, for each experiment, unstained cells were used to establish FSC and SSC signal acquisition thresholds. The minimum threshold was established for only FSC. Fluorescence thresholds for each dye (DAPI, JC-D7, and SYTO®62) were achieved by independent staining to determine the positive and negative limits for each. To avoid biases resulting from contamination or chemical interactions, we analysed each staining buffer alone, with each fluorochrome, or with a combination of dyes.

### 2.9. Observation by epifluorescence microscopy of intracellular polyP after labelling with DAPI

Control counts of the polyP+ cells were carried out by epifluorescence microscopy. Samples were diluted with PBS (between $10^5$ and $10^6$ cells per sample), filtered through black polycarbonate membranes (0.22 µm porosity, 25 mm diameter, GTBP, Millipore®) and stained for 30 min with DAPI (28.5 µM final concentration). Filters were washed with 2 mL of PBS, incubated in the dark at 20 °C with DAPI (2.85 µM, 10 min) or SYTO® 62 (1 µM, 10 min) to visualise cellular DNA. Filters were washed with 2 mL of PBS, dried and mounted with Immersol™ immersion oil (refractive index = 1.518, Zeiss). It should be noted that in order to use labelling conditions similar to those used in cytometry (i.e. double labelling polyp-DAPI and DNA-SYTO®62), we used double labelling in epifluorescence microscopy (polyP-DAPI and DNA-DAPI). However, single labelling with a high concentration (30 min, 28.5 µM final concentration) of DAPI would have been sufficient.

Cells were imaged using a Zeiss™ Axio Imager 2 microscope equipped with a FLUO COLIBRI 5 source with the following light-emitting diodes: UV (385/30 nm), blue (469/38 nm), green (555/30 nm) and red (631/33 nm). The following bandpass filters were applied 450/50 nm, 525/50 nm and 690/50 nm for blue DAPI (DNA), green DAPI (polyP) and SYTO®62, respectively. The diode intensity was adjusted as follows: DAPI blue 2%; DAPI green 100%, SYTO®62 100%. PolyP and lipid inclusions are known to emit in the 450 - 650 nm range when excited at 360 nm, but lipid inclusions can be easily distinguished from polyP, as the fluorescence intensity of the former is much lower and fades rapidly within a few seconds (Terashima *et al.*, 2020). Therefore, all photographed images

were exposed to excitation light for at least 1 min prior to imaging in order to detect consistent and long-lasting bright green-yellow fluorescence from the polyP. Between 200 and 1000 cells were counted per sample. Images were captured and processed using the Zen 3.3 blue edition software.

### 2.10. Transmission electron microscopy coupled with energy dispersive X-ray spectrometry (TEM-EDX)

The samples were fixed with a 2 % formaldehyde solution (final concentration) and collected on 400 mesh copper electron microscopy grids covered with a Formvar film (A03X, Pelanne Instruments, Toulouse, France) by centrifugation (18,000 g, 20 min, 14 °C). After drying, the samples were observed and photographed using a JEOL JEM 2100-Plus transmission electron microscope (TEM), operating at 200 KV (JEOL Ltd, Tokyo, Japan), and equipped with a GATAN RIO 9M camera (Gatan Inc., Pleasanton, California, USA). Chemical elements were analysed in TEM (tilted to 20°) by energy dispersive X-ray spectrometry (EDX) using the X-Max 80 $mm^2$ large area SDD Silicon Drift Detector (Oxford Instruments, Abingdon-on-Thames, United Kingdom) equipped with the AZtec software (Oxford Instruments, Abingdon-on-Thames, UK), in point mode or map mode.

### 2.11. Comparison of polyP staining using DAPI or JC-D7 dyes

The samples were stained with JC-D7 for polyP detection (10 µM final concentration, 30 min of incubation in the dark, Angelova *et al.,* 2014) and with SYTO®62 for DNA colocalization (1 µM, 10 min, in the dark). Treatment used as reference was polyP staining with DAPI (28.5 µM final concentration, 30 min in the dark). The FCM analysis and controls were carried out as described in sections 2.6 and 2.8. The fluorescence of the JC-D7-polyP complexes (after excitation at 405 nm) was collected with a 530/30 nm bandpass filter (green fluorescence).

### 2.12. Statistical analyses

Statistical analyses were performed with GraphPad Prism software, version 8.0.1 for Windows (GraphPad Software, La Jolla California, USA). After Shapiro-Wilk normality and Brown-Forsy homoscedasticity tests, the similarities between treatments were evaluated using a one-way ANOVA or a two-way repeated measures ANOVA, with Tukey's post-hoc test to make multiple comparisons between the groups or an unpaired Student t-test. Data were expressed as the mean ± standard deviation of the mean (mean ± SD). Data were expressed as mean ± standard deviation of the mean (mean ± SD). Differences were considered statistically significant if the *p* value was less than 0.05 ($p < 0.05$).

## 3 Results

### 3.1. Microscopy observations of polyP granules and DAPI-polyP complexes in *T. elongata* and RX cells

Transmission electron microscopy observations confirmed the presence of one or more intracellular electron-dense granules within the *T. elongata* cells (Fig.1A). Although the distribution of carbon within the *T. elongata* cells was relatively homogeneous, energy dispersive X-ray spectrometry revealed higher amounts of P and oxygen as well as the presence of monovalent ($Na^+$, $K^+$) and divalent ($Mg^{2+}$) cations in the granules (Fig. 1A, Fig. S2).

Observations by epifluorescence microscopy revealed highly refractive granules that emit fluorescence consistent with that expected for polyP after labelling with DAPI at 28.5 µM (Fig.1B and 1B', Fig.1C and 1C', Fig. S.3). These observations confirm that the RX strain is a low accumulator (Fig.1B and 1B'), whereas the *T. elongata*

280 strain is a high accumulator (Fig. 1C and 1C') of polyP. The counting of DAPI-polyP complexes in epifluorescence
microscopy was used to validate the cytometric data.

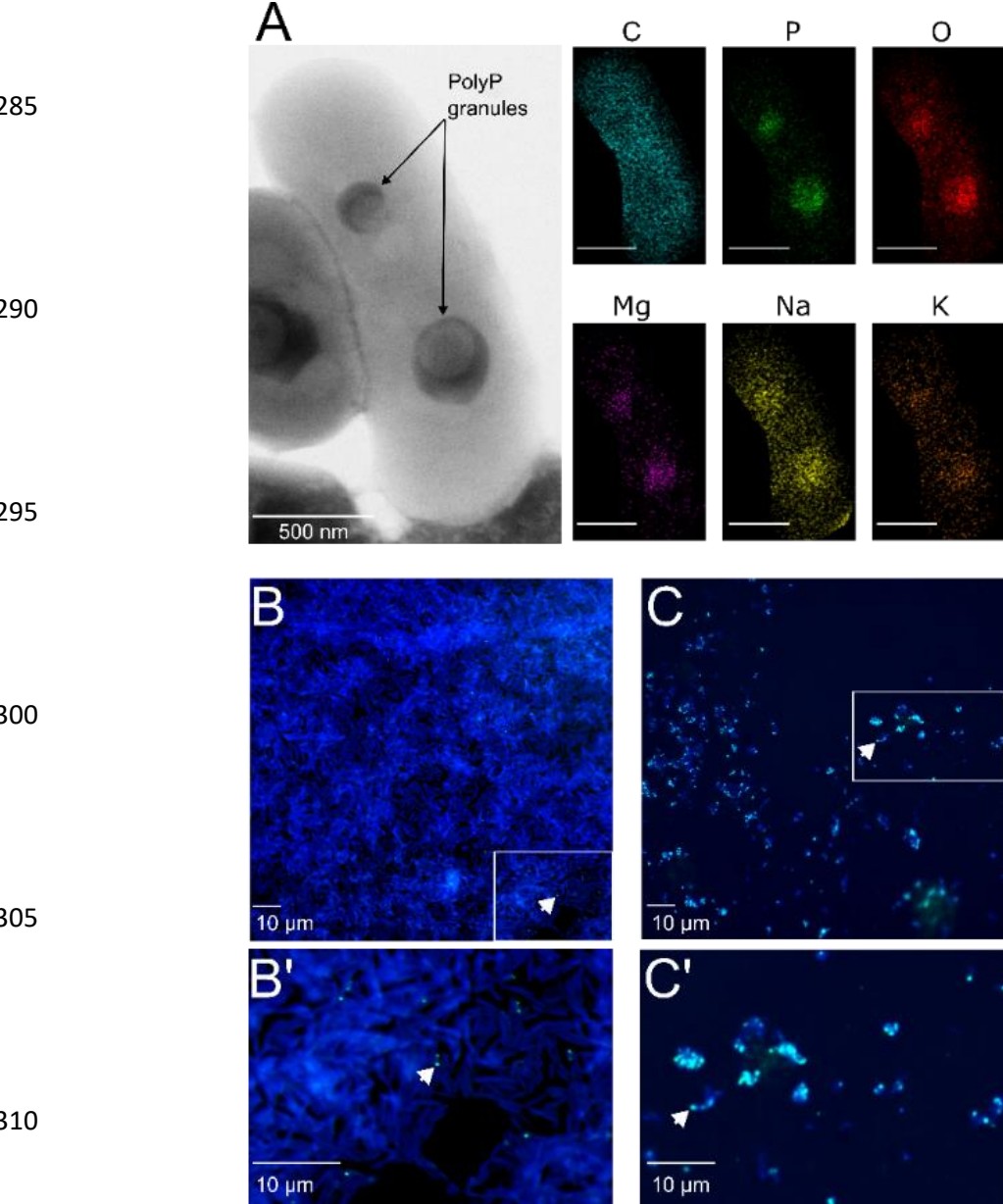

**Figure 1: Transmission electron microscopy coupled with energy dispersive X-ray spectrometry (TEM-EDX), and epifluorescence microscopy images of *T. elongata* and RX cells.**
(**A**) Representative image of two polyphosphate granules in a *Tetrasphaera elongata* Lp2 cell (DSM 14184) with EDX analysis indicating the chemical composition in and out of the granules. The elements shown are C for carbon (false coloured in blue), O for oxygen (false coloured in red), Na for sodium (false coloured in yellow), Mg for magnesium (false coloured in purple), P for phosphorus (false coloured in green), and K for potassium (false coloured in orange). Scale bars represent 500 nm (bottom
left of photographs). (**B**) and (**C**) DAPI-stained images by epifluorescence microscopy of RX and *T. elongata* cells, respectively. DNA and polyP emit a blue and a green-yellow fluorescence (examples are shown by white arrows), respectively. (**B'**) and (**C'**) are zooms of the panels delimited by a white rectangle in images (B) and (C), respectively.


### 3.2. Isotonic buffer for labelling polyP with DAPI in flow cytometry (FCM)

**3.2.1. Staining buffers versus strain populations**

The effect of staining buffers on the *T. elongata* and RX strains was tested in the absence of labelling. The FSC and SSC parameters were analysed after 0, 10- and 20-min incubation of cells in the following buffers: Tris-EDTA, HEPES, PBS and McIlvaine. The Tris-EDTA buffer affected the RX population with the differentiation of a subpopulation (P2) with incubation time (Fig. 2A), suggesting that Tris-EDTA damaged the cellular integrity of

RX cells. Therefore, Tris-EDTA buffer was excluded from further analyses.

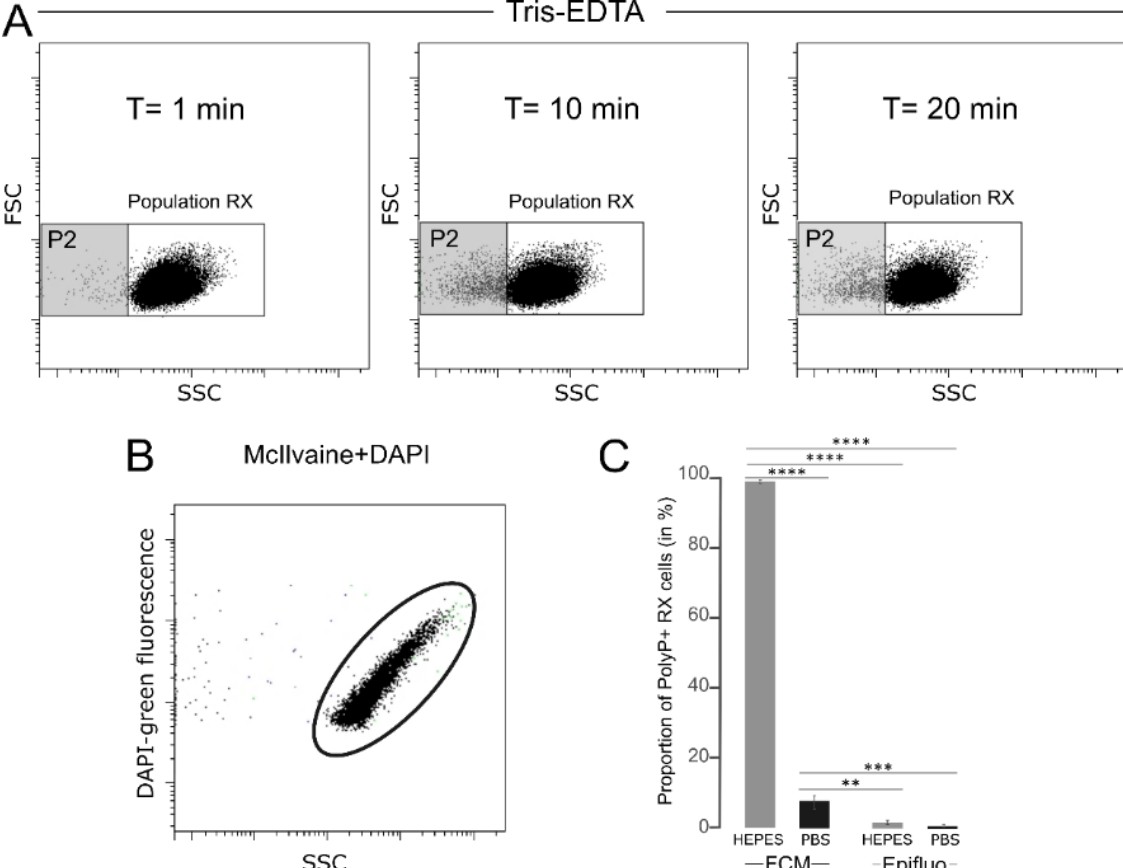

**Figure 2: Tests of different isotonic buffers for the labelling of polyP with DAPI in flow cytometry**
(**A**) Cytograms obtained after T=1 min, T=10 min and T=20 min incubation of unlabelled RX cells in Tris-EDTA buffer. (**B**) Cytogram obtained after DAPI labelling of McIlvaine buffer without cells revealing an artefact signal with green fluorescence.
(**C**) Proportion of polyP+ cells counted by flow cytometry (FCM) or epifluorescence microscopy (Epifluo) after labelling RX cells with DAPI in HEPES or PBS buffer. Significance was determined using one-way ANOVA test and Tukey's post-hoc test for multiple comparisons denoted as follows: $^{*}$p<0.05, $^{**}$p $< 0.001$, $^{***}$p $< 0.0005$, and $^{****}$p $< 0.0001$.
FSC: forward scatter, SSC: Side scatter.

**3.2.2. Staining buffers versus SYTO®62 and DAPI dyes**

The potential interference between dyes and isotonic buffers in the absence of cells was evaluated. No interference was observed between SYTO®62 and the HEPES, PBS, and McIlvaine buffers (Table S.1.). Negative controls were also validated for DAPI in PBS and HEPES buffers (Table S.2). However, artefact labelling was observed between DAPI and McIlvaine buffer (Table S.2), as evidenced by the detection of green fluorescent events in this

cell-free buffer (Fig. 2B). The observed fluorescence was not linked to microbial contamination, as shown after labelling of McIlvaine buffer with SYTO®62 (Table S.1.). Therefore, the McIlvaine buffer was excluded from further analyses.

### 3.2.3. Staining buffers versus labelling performance

Cells from *T. elongata* and RX strain cultures were labelled with the fluorochrome SYTO®62 in PBS or HEPES
buffer. The number of total cells counted using the fluorescence of the SYTO®62-DNA complexes was higher in the HEPES buffer for both strains (Table S.3). The same dye-buffer test was performed after labelling *T. elongata* and RX cell polyP with DAPI. Counts were carried out in FCM and checked by epifluorescence microscopy. The proportion of *T. elongata* cells counted by FCM and containing polyP (polyP+ cells) over the total cells was 82.3 $\pm$ 0.2 % and 87.5 $\pm$ 0.1 % in PBS and HEPES buffer, respectively (Table S.4). The controls performed by
epifluorescence microscopy after labelling the *T. elongata* cells with DAPI confirmed these proportions with polyP+ cells accounting for 87.2 $\pm$ 5.5 % and 92.5 $\pm$ 2.9 %, in PBS and HEPES buffer, respectively (Table S.5). Regardless of the counting method (FCM or epifluorescence microscopy) or buffer (HEPES or PBS), the results for the *T. elongata* strain are consistent (Table S.6). On the other hand, the proportions of polyP+ cells detected in RX cultures diluted in HEPES (99.9 $\pm$ 0.0 %) or PBS (7.2 $\pm$ 2.4%) buffer and counted by FCM were different (Fig.
2C, Table S.7).

Control counts by epifluorescence microscopy (1.9 $\pm$ 1 % and 0.9 $\pm$ 0.5 % for PBS and HEPES, respectively, Table S.8) were significantly different from the FCM (Table S.9), highlighting that the coupling of DAPI staining and FCM is not optimal for low poly-P accumulating organisms. HEPES buffer, which gives an artefact green fluorescence signal for the Gram-negative RX strain, was excluded and PBS buffer, which is a better option than
HEPES buffer, was used for subsequent FCM and FACS analyses. However, PBS buffer is already a compromise and also results in artefact labelling and false positives for the 'low polyP accumulation' control strain.

## 3.3. Permeabilisation and storage conditions

### 3.3.1. Cell permeabilisation

To assess the degree of permeability on the efficiency of polyP labelling, RX and *T. elongata* cells were pretreated with a synthetic detergent, Triton X-100, or a fixative, formaldehyde. As revealed by cytometry data after labelling with SYTO®62, cell incubation in 0.3 % Triton X-100 induced a cell loss of 43.5 $\pm$ 5.2 % and 62.7 $\pm$ 5.2 % for *T. elongata* and RX, respectively (Table S.10). After 1 h of incubation (Day 0), fixation with formaldehyde at a final
concentration of 2 or 4 % had a significant impact on the detection of polyP+ cells for the RX strain compared with the unfixed culture (Fig. 3A, Table S.11 and S.12). No significant difference in the proportion of polyP+ cells was observed for *T. elongata* fixed with 2 % and 4 % formaldehyde compared to unfixed cells (Fig. 3B, Table S.13 and S.14).

### 3.3.2. Cell preservation

PolyP preservation was assessed after formaldehyde fixation at 2 or 4 % by investigating the proportion of polyP+ cells detected after different storage times (t = 2, 7 and 14 days) and temperatures (4 °C, -20 °C, -80 °C) (Table S.11 and S.13). No significant difference was observed for *T. elongata* strain, whatever the formaldehyde

concentration, storage time or temperature (Table S.15). However, storage at -20°C or -80°C is not optimal for the
RX strain, whatever the fixative concentration (Fig. 3C, Table S.16 and S.17). For this Gram-negative strain,
storage at 4°C after fixation with 2% formaldehyde is the most suitable storage condition.

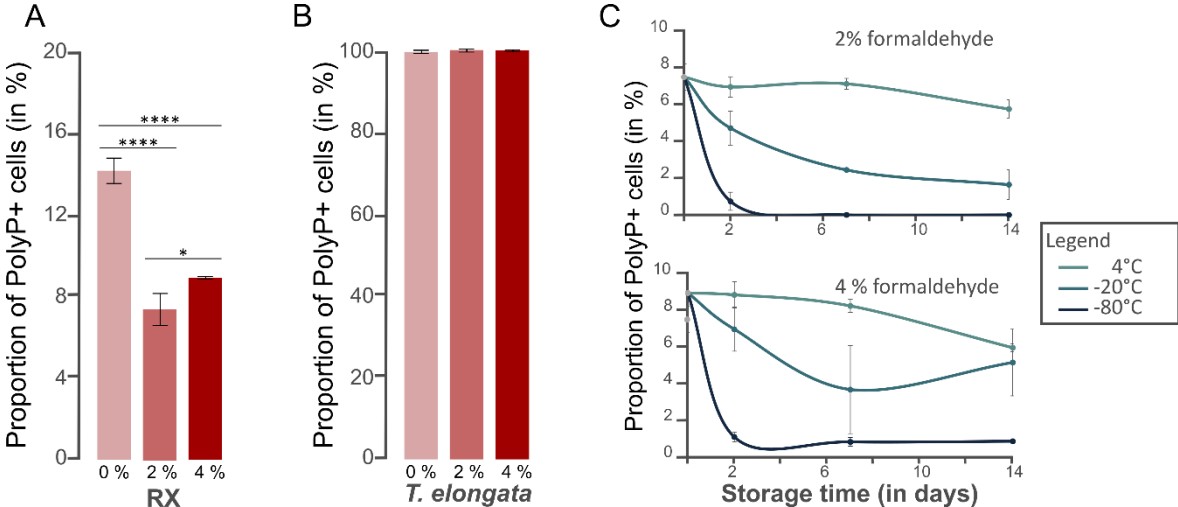

**Figure 3: Preservation of polyP+ as a function of formaldehyde concentration, temperature and storage time**
Proportion of polyP+ cells detected in the **(A)** RX and **(B)** *T. elongata* strain cultures at day 0 without addition of fixative (0
%) and with 2 % and 4 % formaldehyde. Significance was determined using one-way ANOVA test, and Tukey's post-hoc test
for multiple comparisons, denoted as follows: * $p < 0.05$, and **** $p < 0.0001$. **(C)** Proportion of polyP+ cells detected in the RX
strain culture after fixation at 2% (top graph) or 4 % (bottom graph) as a function of storage time (2, 7 and 14 days) and storage
temperature (4 °C, -20 °C, -80 °C).

### 3.4. Validation of the DAPI-labelling protocol for polyP for FACS analyses

First, we prepared a mixed culture of *T. elongata* and RX. After determining the number of RX and *T. elongata*
cells in each strain culture by FCM and SYTO®62 labelling, we mixed them in a 50:50 abundance ratio.
Fluorescence-activated cell sorting (FACS) was performed on this 50:50 mixture (Fig. 4A). We determined the
proportion of polyP+ cells by counting them by epifluorescence microscopy and FCM, after labelling with DAPI,
prior to cell sorting (Fig. 4B). We also carried out these counts using these two approaches after cell sorting in
each of the polyP+ (Fig. 4C and 4E) and polyP- fractions (Fig.4D and 4F). Prior to cell sorting, 36.5 % and 12.6
± 7.5 % of cells were identified as polyP+ in the mixed *T. elongata* + RX culture by FCM and epifluorescence
microscopy, respectively (Fig. 4B, Table S.18).

After cell sorting, $4.5.10^6$ and $4.3.10^6$ cells were collected in the polyP+ and polyP- (i.e. negative green
fluorescence signal regarding the fluorescence intensity limit defined by the controls) fractions, respectively (Fig.
4C and 4D). A strong enrichment of PAB was observed in the polyP+ fraction, as shown by FCM and
epifluorescence microscopy counts (> 80 % of polyP+ cells, Fig. 4E, Table S.18). In contrast, PAB represented
less than 15 % in the polyP- fraction (Fig. 4F, Table S.18).

Cell sorting after labelling of polyP with DAPI and DNA with SYTO®62 was also carried out on a lake water
sample (Table S.19). Cells from the water sample were sorted by FACS and $7.9 \times 10^6$ and $6.3 \times 10^6$ cells were
collected in the polyP+ and polyP- fractions, respectively. PolyP+ cells were counted by epifluorescence
microscopy before and after cell sorting (Table S.19). Prior to cell sorting, the water sample contained $9.7 ± 1.5$
% of polyP+ cells (Table S.19). Target cells enrichment was observed in the polyP+ fraction, with $52 ± 1.8$ % of

polyP+ cells (Table S.19). Although highly significant (p < 0.0001), this enrichment was much less effective than that obtained with the mixture of RX + *T. elongata* strains (Fig. 4).

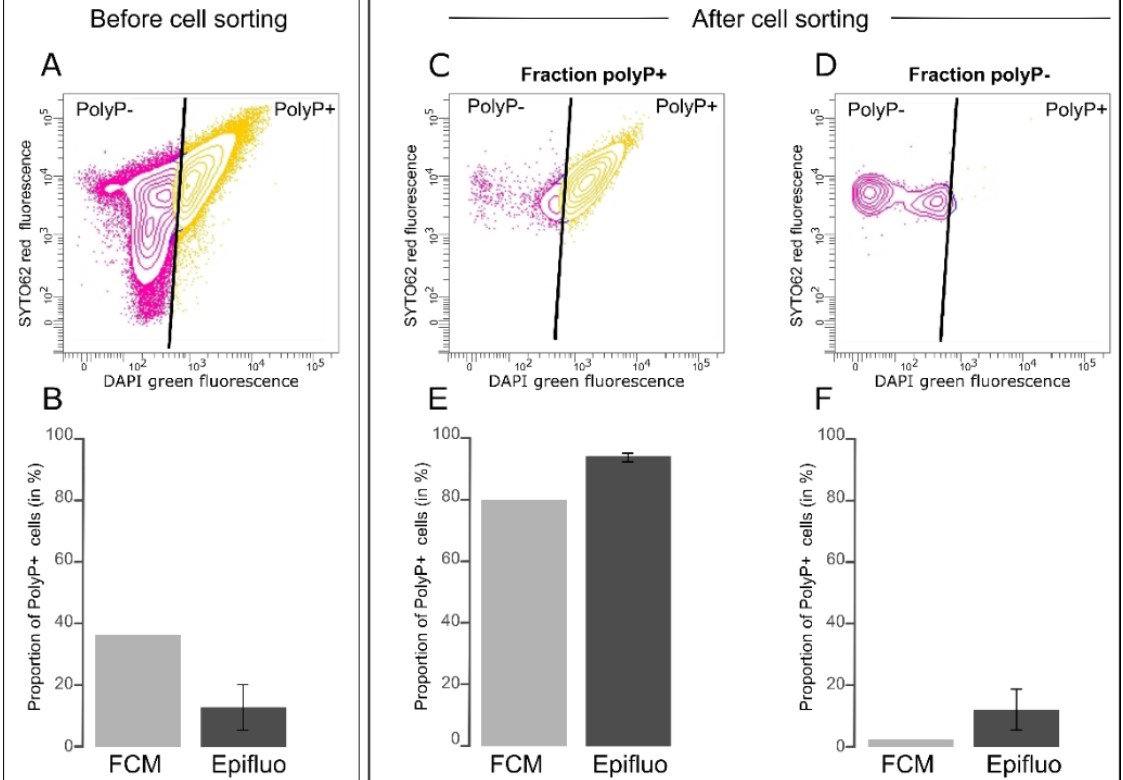

**Figure 4: PAB cell sorting from a mixed culture of *T. elongata* and RX**
(**A**) Cytogram showing the fluorescence of polyP-DAPI complexes (green fluorescence) and the fluorescence of DNA-SYTO®62 complexes (red fluorescence) in the mixed culture of *T. elongata* and RX prior to cell sorting.
(**B**) Proportion of polyP+ cells in the mixed culture of *T. elongata* and RX, labelled with DAPI prior to cell sorting and counted by flow cytometry (FCM) and epifluorescence microscopy (Epifluo).
(**C**) and (**D**) Cytograms showing the fluorescence of polyP-DAPI complexes (green fluorescence) and the fluorescence of DNA-SYTO®62 complexes (red fluorescence) in the (**C**) polyP+ and (**D**) polyP- fraction after cell sorting of the mixed culture of *T. elongata* and RX.
(**E**) and (**F**) Proportion of polyP+ cells in fractions (**C**) polyP+ and (**D**) polyP- after cell sorting of the mixed culture of *T. elongata* and RX and counted by flow cytometry (FCM) and epifluorescence microscopy (Epifluo).
Standard deviations are not shown for FCM because only one sample was counted per fraction.

### 3.5. Counting of PAB from strain cultures and environmental samples using DAPI or JC-D7 labelling

In parallel to the DAPI labelling of polyP, tests were performed on fresh bacterial strain cultures or environmental samples using the dye JC-D7, which is known to be specific for polyP. By staining *T. elongata* cells in HEPES buffer, as recommended by Angelova *et al.* (2014), we found that the green fluorescence intensity of JC-D7 at 525 nm was lower than that of DAPI (Fig.S4). We also performed polyP labelling of cells with JC-D7 in PBS buffer, but this proved to be sub-optimal as it showed weak green fluorescence that was difficult to separate from the fluorescence of the negative controls (Fig. S.4).

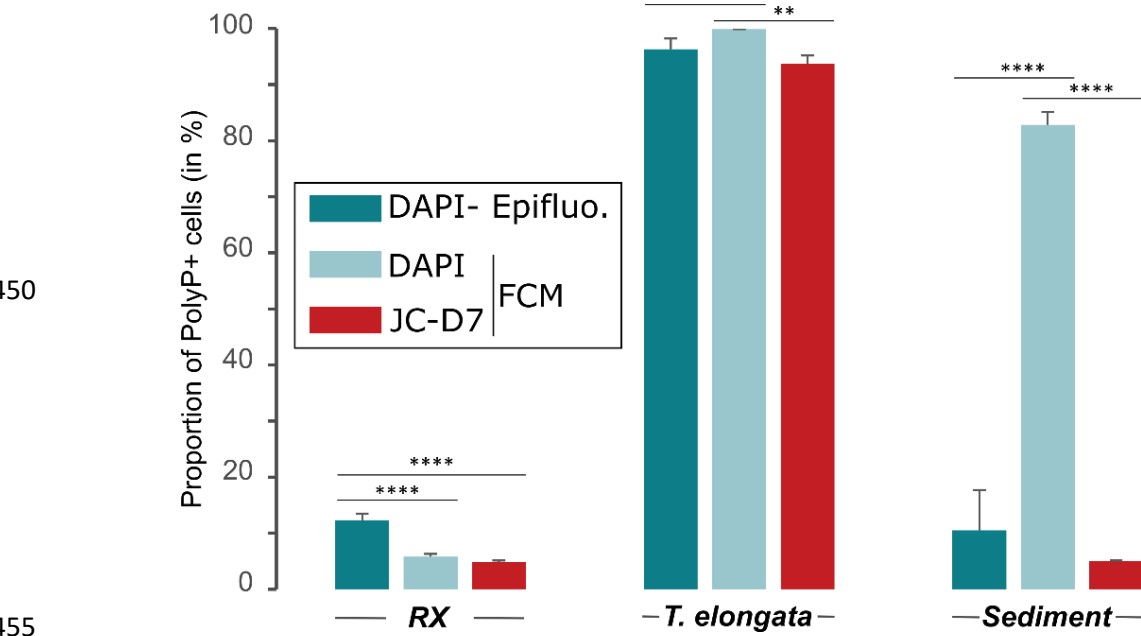

**Figure 5: Comparison of JC-D7 and DAPI labelling for PAB detection**

 Proportion of polyP+ cells, after PAB labelling with DAPI and SYTO®62 or with JC-D7 and SYTO®62. Cells were counted by flow cytometry (FCM) or epifluorescence microscopy (Epifluo). Significance was determined using one-way ANOVA test and Tukey's post-hoc test for multiple comparisons, denoted as follows: *$p < 0.05$, **$p < 0.001$, ***$p < 0.0005$, and ****$p < 0.0001$.

Culture samples of *T. elongata* and RX strains and lake sediments were labelled with DAPI or JC-D7 for polyP
 and SYTO®62 for DNA. The proportion of polyP+ cells was counted in FCM and control counts, on the same samples, were carried out using epifluorescence microscopy (Table S.20 to S.22). For RX and *T. elongata* strains, as observed previously (Fig. 4), the DAPI-labelled polyP+ cell counts, although showing significant differences between FCM and epifluorescence microscopy, were in similar proportions (Fig. 5). For the *T. elongata* strain, the proportion of polyP+ cells after polyP labelling with JC-D7 fluorochrome ($93.7 \pm 1.5$ %) was not significantly
 different from that determined by epifluorescence microscopy ($96.3 \pm 1.9$ %, Fig. 5, Table S.20). For RX strain, the proportions of polyP+ determined by FCM after labelling with JC-D7 ($4.8 \pm 0.3$ %) or DAPI ($5.8 \pm 0.5$ %) were not significantly different (Fig. 5, Table S.21). These proportions were significantly lower than those obtained after counting by epifluorescence microscopy ($12.3 \pm 1.2$ %, Fig. 5, Table S.21). For the lake sediment sample, DAPI and JC-D7 fluorochromes led to a very different detection of polyP+ cells by FCM ($82.8 \pm 2.3$ % and $5 \pm$
 $0.1$ % polyP+ cells, respectively; Fig. 5, Table S.22). Control counts by epifluorescence microscopy ($10.5 \pm 7.2$ %) were not significantly different from FCM counts after labelling with JC-D7 (Fig. 5, Table S.22).
Comparative polyP labelling assays using the same methodology were extended to Gram-negative strains belonging to *Acinetobacter lwoffii, Flavobacterium sp.*, *Pseudomonas sp.*, *Stenotrophomonas rhizophila* and the Gram-positive strain *Microbacterium hydrocarbonoxydans* DSM 16089. A culture of *T. elongata* was also
 included in this series of experiments as a positive control (Fig. 6). In the *T.elongata* culture, the proportion of polyP+ cells counted by FCM after labelling with DAPI or JC-D7 was not significantly different, but was overestimated as shown by the epifluorescence microscopy control counts (Fig. 6a and 6b, Table S.23 and S.24),.

For the other strains, no polyP+ cells were visualised by epifluorescence microscopy (data not shown). These observations are consistent with the FCM counts after labelling with DAPI or JC-D7 (Fig. 6c to 6g, Table S.23
485    and S.24)

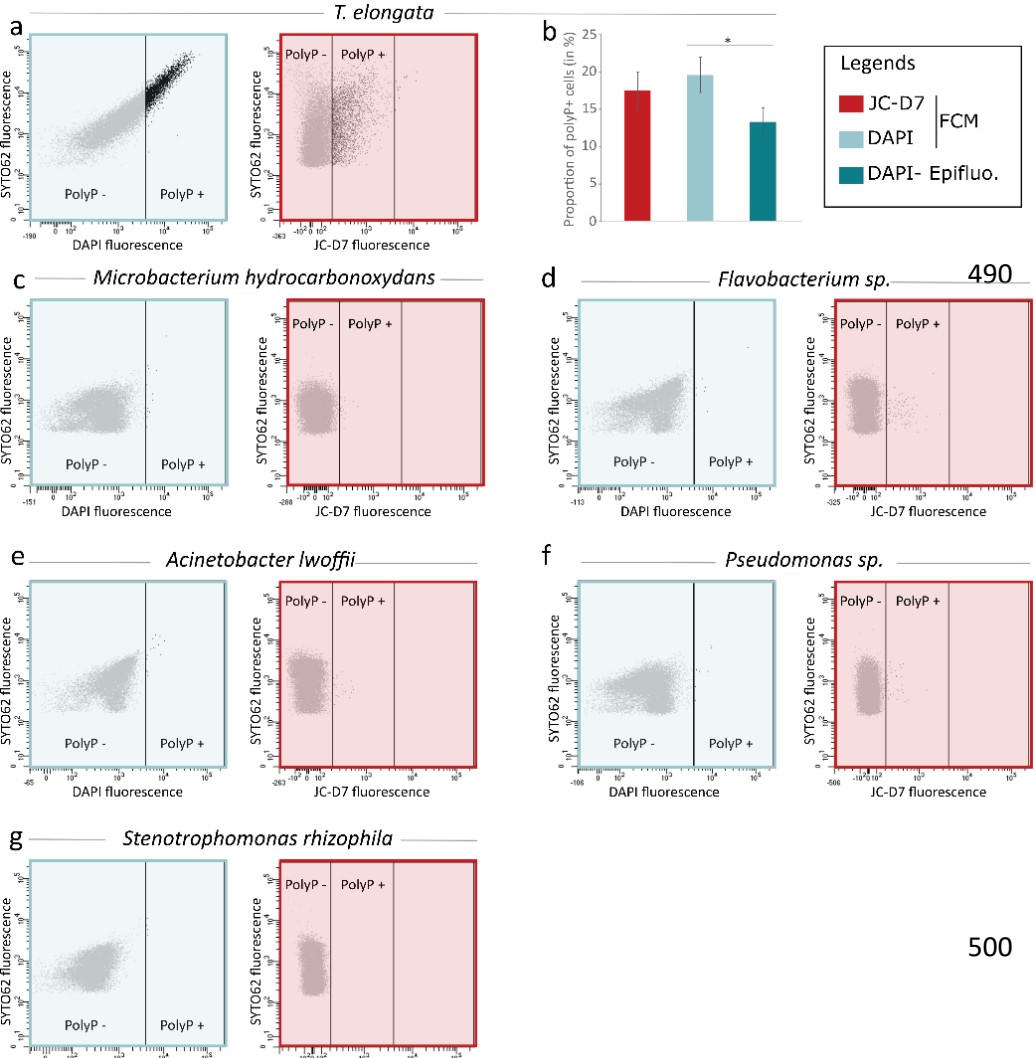

**Figure 6: JC-D7 and DAPI labelling tests for the detection of polyP+ cells for different bacterial strains.**
**(a)** Cytograms showing the fluorescence of polyP-DAPI or polyP-JC-D7 complexes and the fluorescence of DNA-SYTO®62
505    complexes in the culture of *T. elongata*. **(b)** Proportions of polyP+ cells for *T. elongata* after PAB labelling with DAPI or with JC-D7. Cells were counted by flow cytometry (FCM) or epifluorescence microscopy (Epifluo). Significance was determined using one-way ANOVA test and Tukey's post-hoc test for multiple comparisons, denoted as follows: *p < 0.05.
**(b)** to **(h)** Cytograms showing the fluorescence of polyP-DAPI (blue box) or polyP-JC-D7 complexes (red box) and the fluorescence of DNA-SYTO®62 complexes in the cultures of **(c)** *Microbacterium hydrocarbonoxydans* DSM 16089, **(d)**
510    *Flavobacterium sp*., **(e)** *Acinetobacter lwoffii*, **(f)** *Pseudomonas sp*., **(g)** *Stenotrophomonas rhizophila*.

## 3.6. Ecological relevance of flow cytometric detection of polyP/JC-D7 complex fluorescence

We tested the ecological relevance of coupling JC-D7-polyP labelling with flow cytometry by enumerating PAB
515    in soils, sampled in triplicate in conventional (n = 3) or organic (n = 3) farming parcels with contrasting concentrations of available orthophosphate (Table S.25). Concentrations of bioavailable orthophosphate averaged 72.6 ± 34.2 µg/g dry soil and 34.9 ± 6.7 µg/g dry soil in conventional and organic farming parcels, respectively

(Fig. 7, Table S.25). The relative proportions of PAB were on average 3.7 ± 1.4% and 0.4 ± 0.2% in the conventional and organic parcels, respectively (Fig. 7, Table S.25). A significant positive correlation was observed between the proportion of PAB and the concentration of bioavailable P (r = 0.69, p = 0.003 ; Fig. 7).

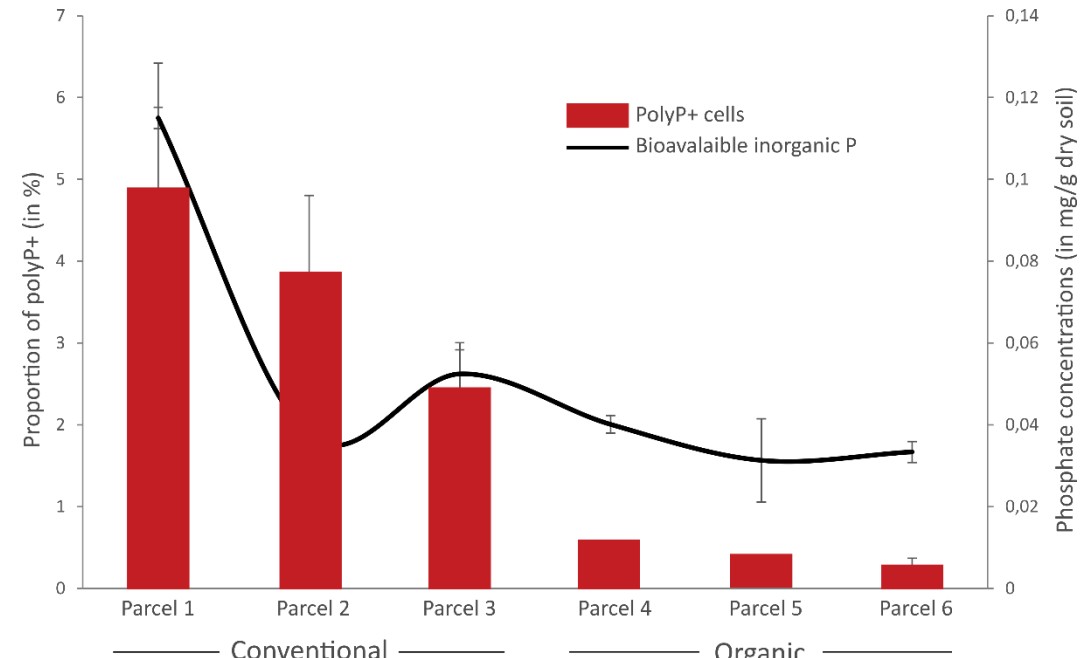

**Figure 7: Proportion of polyphosphate-accumulating bacteria (PAB) and concentrations of bioavailable orthophosphate in conventional or organic farming parcels.**
The proportion of polyP+ cells was determined by flow cytometry as the number of cells showing a positive green fluorescence signal after labelling with JC-D7 compared to the total number of cells after labelling with SYTO®62. Concentrations of bioavailable inorganic orthophosphates were estimated by the Olsen extraction method. Error bars are not shown for polyP+ cells for parcels 4 and 5 as only two replicate samples were analysed.

## 4   Discussion

### 4.1. Optimisation of polyP labelling with DAPI

The coupling of specific polyP detection to FCM has been used in a very small number of studies that have applied standardised conditions previously defined in the literature, for example, for polyP detection by epifluorescence microscopy (e.g. Mesquita *et al.,* 2013; Voronkov *et al*., 2019). In this study, we tested several variables likely to affect the efficiency and reliability of PAB detection in the environment. These variables, which include the choice of buffer, fluorochrome, fixative concentration or storage conditions, may seem trivial but they are essential for defining future standardised protocols and should be acquired as a priority. This will enable the scientific community to avoid pitfalls and save time and effort in carrying out tedious analyses.

Our data show that the parameters for labelling polyP with DAPI are not necessarily transposable from *ad hoc* conditions adapted to epifluorescence microscopy and that specific adaptation of the labelling methods is required for the FCM approach. For example, McIlvaine buffer, which has been identified to significantly enhance DAPI-polyP labelling in epifluorescence microscopy (Mukherjee and Ray, 2015), causes events which size and structure mimic nonviable cells. These observed artefact signals, associated with McIlvaine-DAPI interactions, whose physicochemical origin has not been identified, are highlighted by FCM.

In contrast, the intense fluorescence of green DAPI when cells were labelled in HEPES buffer suggests that the latter is an optimal solvent for the detection of polyP by flow cytometry. However, this fluorescence detection was artefactual and led to an overestimation of the proportion of polyP+ cells within the Gram-negative RX population as revealed by epifluorescence microscopy observations on the same samples. Based on the tests performed in this study, we conclude that the choice of buffer for labelling polyP with DAPI should be PBS, which is also a frequently used buffer for labelling DNA with this fluorochrome. However, the PBS buffer is a compromise as false positives were also recorded for the low polyP accumulation control strain.

In addition, we show that membrane permeabilisation steps are not necessary for the detection of polyP by DAPI labelling in FCM. On the contrary, significant losses were recorded in terms of the total number of events and proportions of polyP+ cells with two compounds, a detergent (Triton X100) and a fixative (formaldehyde), which have different modes of action but both permeabilise cell membranes. The effect of fixation on the cytometric analysis of bacteria is well known in ecology (e.g. Kamiya *et al.*, 2007; Troussellier *et al.*, 1995) and a differential effect of fixative between Gram-positive and Gram-negative bacteria has already been observed (Liu *et al.,* 2012). In order to reconcile this methodological approach with the constraints of environmental studies, which often do not allow samples to be analysed immediately, the conservation of polyP was considered. The bias induced immediately after formaldehyde fixation for the detection of PAB did not increase significantly after samples were stored for 14 days at 4 °C for RX and for all storage temperatures for *T. elongata*. Consequently, we recommend storage, after fixation with 2 % formaldehyde, at 4 °C. This temperature avoids freeze-thaw cycles that could damage cells during repeated analyses of the same sample. It is also compatible with transporting samples from the sampling site under *ad hoc* conditions.

We also simultaneously applied DNA-SYTO®62 labelling and polyP-DAPI labelling. This double staining, which has not been used in previous studies (e.g. Günther *et al.,* 2009; Terashima *et al.,* 2020), enables the separation of PAB from common autofluorescent contaminants such as aggregates or organic matter. In addition, the choice of SYTO®62 avoids interference with the metachromatic properties of DAPI by using a fluorochrome whose emission spectra are perfectly separable from the green and blue fluorescence of DAPI (Fig. S1).

### 4.2. Is polyP labelling with DAPI really applicable to FCM in environmental samples?

The number of polyP cells counted by FCM was checked by epifluorescence microscopy, a standard method for quantification and visualisation of PAB (Serafim *et al.*, 2002). For almost all the data obtained, the observations by FCM and epifluorescence are significantly different, and especially in natural samples, the differences were striking. Should FCM therefore be recommended for the detection of DAPI-labelled polyP?

DAPI is an inexpensive dye for which most epifluorescence microscopes and flow cytometers have combinations of excitation and emission filters compatible with the detection of its blue or green fluorescence (e.g. Tarayre *et al.,* 2016). However, DAPI is also a nonspecific polyP dye. In addition to labelling DNA (blue fluorescence), it also interacts with lipids, displaying metachromatic properties similar to those of polyP (Serafim *et al.,* 2002). Although the DAPI-lipid fluorescence is short-lived, with respect to the speed of flow cytometry analysis, it cannot be ignored whereas in microscopy the longer exposure times for counting make it possible to avoid the artefact counting of lipids.

It should also be noted that FCM and epifluorescence microscopy are different methods, each with its own advantages and pitfalls. The main advantage of flow cytometry is its counting speed, which makes it possible to

establish the proportions of PAB in a large sample of cells from the targeted microbial community or population. Collecting hundreds of thousands of events in minutes offers incredible opportunities, but also bears certain risks, for example in defining the gating strategy. Microscopy, in turn, allows visualization of the research material but does not count many cells (generally up to 400; Kepner and Pratt, 1994) and often depends on the experimenter. Since the basic assumption of most statistical tests is that samples are randomly and independently selected, the approach of randomly selecting fields of view or squares of an ocular graticule to count bacteria is generally practised in epifluorescence microscopy (Kirchman *et al.,* 1982). However, this approach has been shown to produce a significant statistical bias, contributed for example to 60-80 % of the total variance in bacteria count data for seawater samples (Kirchman *et al.,* 1982). This could be explained by the fact that the actual distribution of bacterial cells in environmental samples may not be random and thus contribute to the overestimation or underestimation of total bacterial abundance.

Despite the substantial variation and subjectivity associated with microscopy and flow cytometry, both methods revealed, with appropriate labelling (e.g. appropriate staining buffer), fairly congruent proportions of polyP+ cells for high polyP accumulators in homogeneous samples, i.e. strain culture samples. The method combining FCM and DAPI labelling of polyP can therefore be applied to such 'simple' matrices, for example to screen a heterogeneous pool of PAB cells or a single PAB strain for gene-phenotype linkage analyses. The use of flow cytometry combined with DAPI labelling of polyP to count PAB in complex matrices such as sediment or freshwater samples can be used for comparative purposes, but cannot be recommended as a sole method due to high risk of overestimation.

Although the FCM-DAPI labelling approach is not reliable for PAB counts in complex samples, our results confirm that PAB can be enriched from the environmental microbial community using FACS and DAPI-polyP labelling, which opens up interesting prospects. Terashima *et al.* (2020) have demonstrated that ~70% of cells remain viable after DAPI staining, enabling effective enrichment and isolation of PAB after FACS-based phenotype screening. This 'targeted enrichment' may also be combined with metagenomics (Thompson *et al.,* 2013) to improve recovery of metagenome-assembled genomes of environmental PAB. In light of our data, we acknowledge that this approach will be more effective in the enrichment and isolation of PAB that accumulate large amounts of polyP.

**4.3 The polyP specific fluorochrome JC-D7 as a prospective marker for complex environmental samples**

Considering that DAPI labelling is not suitable for enumerating PAB in environmental samples due to the non-specificity of this fluorochrome, the best option would therefore be to use markers specific to polyP. This approach was advocated by Günther *et al.* (2009) who proposed using the bright green fluorescence of the antibiotic tetracycline when it complexes divalent cations acting as a countercharge in polyphosphate granules. Prior to the study presented here, we had carried out assays by labelling cultured strains and/or environmental samples with tetracycline but had not obtained convincing results. Therefore, an exploratory analysis was performed to determine the potential of a heterocycle of the benzimidazole family for the detection of PAB by flow cytometry. Prior to our study, the dye JC-D7, identified as specific for polyP, had never been used to target PAB, but only to stain polyP in living eukaryotic cells and tissues (Angelova *et al.,* 2014). The JC-D7 dye was shown to allow for specific labelling of synthetic polyP *in vitro* as well as endogenous polyP in living mammalian cells which have a dramatically lower abundance of polyP compared to microorganisms. In addition, this probe demonstrated high

selectivity for the labelling of polyP that was not sensitive to a number of ubiquitous organic polyphosphates, notably RNA (Angelova *et al*., 2014).

Although we identified artefactual signals during polyP-DAPI labelling in HEPES buffer for the RX strain (Fig. 2), polyP-JC-D7 labelling was performed in this buffer as recommended by Angelova *et al*. 2014. Indeed, the fluorescence of JC-D7 in PBS buffer was not optimal, as it showed a weak green fluorescence that was difficult to separate from the fluorescence of the negative controls (Fig. S4). On the other hand, we did not observe any artefactual signals for the Gram-positive and Gram-negative strains tested when polyP were stained with JC-D7

in HEPES (Fig. 6). The artefacts observed during polyP-DAPI labelling do not appear to be related to erroneous measurements of fluorescence parameters assessed by FCM due to changes in the bacterial membrane of Gram-negative cells caused by the composition of HEPES (e.g. absence of divalent cations, Tomasek *et al*., 2018).

We show that JC-D7 is a promising marker for the enumeration of polyP-accumulating strains. For example, data obtained on *T. elongata* show that this JC-D7/FCM approach particularly reveals the variation in the polyP content

of microorganisms (e.g. $93.7 \pm 1.5$ % and $17.5 \pm 2.4$ % of polyp+ cells in *T. elongata* culture; Fig. 5 and Fig. 6), a common feature of PAB (Fleming, 1992). The data obtained in the present study show that the cell counts obtained with JC-D7 in FCM on environmental samples were not statistically different from those obtained by epifluorescence microscopy after labelling with DAPI (Fig. 5). JC-D7 looks promising for the use in natural samples and may prove useful in many environmental studies. In this context, we carried out PAB counts in soils

with different concentrations of bioavailable orthophosphates. Our data, which show a significant positive correlation between the proportion of PAB and bioavailable orthophosphate concentrations (Fig.7), are in line with that would be expected in soils (Fleming, 1992), underlining the relevance of this method for obtaining ecological data. Our results show that the JC-D7/FCM coupling could provide a direct method to detect the presence of polyP in the biomass of soil microorganisms and to enumerate PAB, whereas current methods in these ecosystems only

have the potential to indirectly indicate the presence of polyP, i.e. growth on P-free media and the associated measurement of net changes in the stoichiometry of the microbial biomass, colour of the microbial suspension stained by the Neisser method (Capek *et al.,* 2024).

Our work is preliminary and the relevance of JC-D7 for the study of PAB needs to be confirmed in further studies. Since the publication of our preprint in Biogeosciences Discussions on May 2, 2024, two articles reporting the use

of JC-D7 for polyP quantification in microbial samples have been published and have also shown JC-D7 to be very promising and specific. The results of assays using JC-D7 to perform a semi-quantitative polyP assessment in yeast extracts suggest that staining with this dye provides a robust and sensitive method for detecting polyP (Deidert *et al.,* 2024). These authors showed that the fluorescence of JC-D7 was unaffected by inorganic phosphate up to 50 mM and only slightly affected by other parameters, such as pH and temperature. They also found that

trace elements (e.g. $FeSO_4$) and toxic mineral salts (e.g. $PbNO_3$) decreased polyP-induced JC-D7 fluorescence, limiting its applicability to samples containing polyP-metal complexes (Deidert *et al.,* 2024). The JC-D7 dye has also been used to quantify polyP in planktonic environmental samples because, as in our study, the authors found that DAPI fluorescence greatly overestimated polyP due to interference (Yang *et al.* 2024). These authors conluded from their study that polyP quantification using JC-D7 fluorescence overcomes the interference encountered by

the DAPI method and provides an efficient, convenient and accurate method of quantifying polyP in planktonic samples in both culture and natural samples (Yang *et al*., 2024).

Our study is a first step towards the quantification and enrichment of PAB in natural samples using the JC-D7 dye. The next important step will be to demonstrate the specificity of the JC-D7 labelling of PAB by using fluorescence microscopy to visualise the JC-D7 fluorescence in combination with, for example, scanning-electron microscopy combined with energy dispersive spectroscopy (SEM-EDS) to perform co-localisation analysis. The development of a standardised protocol to quantify PAB after staining polyP with the JC-D7 probe using epifluorescence microscopy will require further development, particularly due to the weaker fluorescence of JC-D7 compared to DAPI that we demonstrated in our study. Technical advances in cytometry methods, particularly spectral cytometry, may also allow co-labelling of DAPI and JC-D7. This approach is currently not possible using conventional cytometry because the spectra of polyP-DAPI and polyP-JC-D7 overlap and the detection wavelengths are very close (520 nm and 530 nm respectively).

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

**Author contribution**

CB, HB, CCB, KB, F S-P, ED and ACL designed the work. CB, HB, and ACL designed the experiments. ED was responsible for the research project (ANR Phostore). CB, HB, CCB, JC, JA, IB and ACL carried out the experiments. ACL, CB and HB wrote the manuscript and all the authors revised it.


**Competing interest**

The authors declare that they have no conflict of interest

**Acknowledgments**

Clémentin Bouquet was supported by PhD fellowship from the French Ministry of Education and Research. Cécile
Bidaud was supported by the Ecole Doctorale FIRE-Programme Bettencourt. The authors would also like to thank Christopher Lefevre (BIAM, UMR 7265, CEA Cadarache) and Nicolas Menguy (IMPMC, UMR CNRS 7590, Sorbonne university). This work was supported by the Agence Nationale de la Recherche (PHOSTORE: ANR-19-CE01-0005). Young-Tae Tchang was supported by the National Research Foundation of Korea (NRF) grant funded by the Korea government (MSIT) (2023R1A2C300453411) and by the Glocal University 30 project
(Molecular Imaging Center, POSTECH).


**Legends of figures**


**Figure 1: Transmission electron microscopy coupled with energy dispersive X-ray spectrometry (TEM-EDX), and epifluorescence microscopy images of *T. elongata* and RX cells.**

**(A)** Representative image of two polyphosphate granules in a *Tetrasphaera elongata* Lp2 cell (DSM 14184) with EDX analysis indicating the chemical composition in and out of the granules. The elements shown are C for carbon

(false coloured in blue), O for oxygen (false coloured in red), Na for sodium (false coloured in yellow), Mg for magnesium (false coloured in purple), P for phosphorus (false coloured in green), and K for potassium (false coloured in orange). Scale bars represent 500 nm (bottom left of photographs). **(B)** and **(C)** DAPI-stained images by epifluorescence microscopy of RX and *T. elongata* cells, respectively. DNA and polyP emit a blue and a green-yellow fluorescence (examples are shown by white arrows), respectively. **(B')** and **(C')** are zooms of the panels

delimited by a white rectangle in images (B) and (C), respectively.

**Figure 2: Tests of different isotonic buffers for the labelling of polyP with DAPI in flow cytometry**

**(A)** Cytograms obtained after T=1 min, T=10 min and T=20 min incubation of unlabelled RX cells in Tris-EDTA buffer. **(B)** Cytogram obtained after DAPI labelling of McIlvaine buffer without cells revealing an artefact signal

with green fluorescence. **(C)** Proportion of polyP+ cells counted by flow cytometry (FCM) or epifluorescence microscopy (Epifluo) after labelling RX cells with DAPI in HEPES or PBS buffer. Significance was determined using one-way ANOVA test and Tukey's post-hoc test for multiple comparisons denoted as follows: $*$ p<0.05, $**$ $p < 0.001$, $***$ $p < 0.0005$, and $****$ $p < 0.0001$.

FSC: forward scatter, SSC: Side scatter.


**Figure 3: Preservation of PolyP+ as a function of formaldehyde concentration, temperature and storage time**

Proportion of polyP+ cells detected in the **(A)** RX and **(B)** *T. elongata* strain cultures at day 0 without addition of fixative (0 %) and with 2 % and 4 % formaldehyde. Significance was determined using one-way ANOVA test,

and Tukey's post-hoc test for multiple comparisons, denoted as follows: $*$ $p < 0.05$, and $****$ $p < 0.0001$. **(C)** Proportion of polyP+ cells detected in the RX strain culture after fixation at 2% (top graph) or 4 % (bottom graph) as a function of storage time (2, 7 and 14 days) and storage temperature (4 °C, -20 °C, -80 °C).

**Figure 4: PAB cell sorting from a mixed culture of *T. elongata* and RX**

**(A)** Cytogram showing the fluorescence of polyP-DAPI complexes (green fluorescence) and the fluorescence of DNA-SYTO®62 complexes (red fluorescence) in the mixed culture of *T. elongata* and RX prior to cell sorting.

**(B)** Proportion of polyP+ cells in the mixed culture of *T. elongata* and RX, labelled with DAPI prior to cell sorting and counted by flow cytometry (FCM) and epifluorescence microscopy (Epifluo).

**(C)** and **(D)** Cytograms showing the fluorescence of polyP-DAPI complexes (green fluorescence) and the

fluorescence of DNA-SYTO®62 complexes (red fluorescence) in the **(C)** polyP+ and **(D)** polyP- fraction after cell sorting of the mixed culture of *T. elongata* and RX.

**(E)** and **(F)** Proportion of polyP+ cells in fractions **(C)** polyP+ and **(D)** polyP- after cell sorting of the mixed culture of *T. elongata* and RX and counted by flow cytometry (FCM) and epifluorescence microscopy (Epifluo). Standard deviations are not shown for FCM because only one sample was counted per fraction.


**Figure 5: Comparison of JC-D7 and DAPI labelling for PAB detection**

Proportion of polyP+ cells, after PAB labelling with DAPI and SYTO®62 or with JC-D7 and SYTO®62. Cells were counted by flow cytometry (FCM) or epifluorescence microscopy (Epifluo). Significance was determined using one-way ANOVA test and Tukey's post-hoc test for multiple comparisons, denoted as follows: $^*$p < 890 0.05, $^{**}$p < 0.001, $^{***}$p < 0.0005, and $^{****}$p < 0.0001.

**Figure 6: JC-D7 and DAPI labelling tests for the detection of polyP+ cells for different bacterial strains.**

**(c)** Cytograms showing the fluorescence of polyP-DAPI or polyP-JC-D7 complexes and the fluorescence of DNA-SYTO®62 complexes in the culture of *T. elongata*. **(b)** Proportions of polyP+ cells for *T. elongata* after PAB 895 labelling with DAPI or with JC-D7. Cells were counted by flow cytometry (FCM) or epifluorescence microscopy (Epifluo). Significance was determined using one-way ANOVA test and Tukey's post-hoc test for multiple comparisons, denoted as follows: $^*$p < 0.05.

**(c)** to **(h)** Cytograms showing the fluorescence of polyP-DAPI (blue box) or polyP-JC-D7 complexes (red box) and the fluorescence of DNA-SYTO®62 complexes in the cultures of **(c)** *Microbacterium hydrocarbonoxydans* 900 DSM 16089, **(d)** *Flavobacterium sp*., **(e)** *Acinetobacter lwoffii,* **(f)** *Pseudomonas sp*., **(g)** *Stenotrophomonas rhizophila*.

**Figure 7: Proportion of polyphosphate-accumulating bacteria (PAB) and concentrations of bioavailable orthophosphate in conventional or organic farming parcels.**

The proportion of polyP+ cells was determined by flow cytometry as the number of cells showing a positive green fluorescence signal after labelling with JC-D7 compared to the total number of cells after labelling with SYTO®62. Concentrations of bioavailable inorganic orthophosphates were estimated by the Olsen extraction method. Error bars are not shown for polyP+ cells for parcels 4 and 5 as only two replicate samples were analysed.