# Peer review of "Technical note: Flow cytometry assays for the detection, counting, and cell-sorting of polyphosphate-accumulating bacteria"

_EGUsphere, 2024_

## Author Response (AR1)

**Responses to reviewer 1**

We would like to thank the anonymous reviewer for the careful reading and very interesting comments. As this reviewer indicates, coupling polyP detection with a high-throughput approach opens up highly interesting avenues of research. This coupling has already been tested in a small number of previous studies, but a general framework leading to a generalisable protocol is currently lacking towards 'routine' use. This is the purpose of the present study. While our results are very encouraging, the community must continue its efforts to ensure the robustness of the PolyP detection by flow cytometry, in particular by using specific polyP dyes. To this end, we will follow the recommendations of reviewer 1 and moderate our assertions in the corrected manuscript by further highlighting that we show that coupling polyP detection with DAPI labelling in cytometry is not relevant for complex natural samples.

In order not to make the manuscript too unwieldy, we have chosen not to present certain figures and data, as they do not provide any additional information to that given in the text. It is true, however, that this decision can be embarrassing for the reader and leave some doubt as to the validity of the results. Therefore, in agreement with reviewer 1, all data cited in the manuscript and not included in the main document will be presented in the form of supplementary figures and/or made available on the Figshare repository.

You will find below the answers posted on EGUSPHERE (in black) as well as the corrections made in the revised manuscript (in red).

**Responses to general comments**

**GC1- Reviewer comment** : « *The authors grew a known PolyP-accumulating strain (T. elongata) and used a second strain as a control (their own isolate), that still showed "low" levels of PolyP accumulation. This is one of the first flaws of the experimental approach, as the control strain did in fact also accumulate polyphosphates.* »

**Author's response :** PolyP occurs as a ubiquitous biopolymer in representatives of all kingdoms of living organisms and every cell type in nature (*LorenzoOrts et al. 2020, New Phytologist ; Akbari et al. 2021, Microbial Biotechnology*). In all prokaryotic cells, polyPs are found in granules. This storage of polyPs can be exacerbated in certain bacteria, which accumulate these polymers to intracellular concentrations in the millimolar range. PolyP represent up to 20% of cell dry weight (Martin *et al*. 2014). This is the case for *T. elongata*, a bacterium that hyperaccumulates polyP and which we used as a « positive control » in this study. As there is no true negative control, prior to this study we screened our bacterial strain library in the laboratory to identify the strain with the lowest accumulation that we used as a « negative control ». We agree that these terms may be ambiguous. To remove any ambiguity, we propose to use the terms « high accumulator » and « low-accumulator » in the revised manuscript to describe the *T. elongata* and RX strains, respectively.

This is corrected in the revised manuscript: page 1-line 24; page 3-lines 100-103; page 8-lines 290.

**GC2- Reviewer comment** : « *The authors tested a multitude of conditions for staining and storage of cells, although it does not become clear why all these conditions were tested in the first place.* »

**Author's response :** The coupling of specific polyP detection to flow cytometry has been used in a very small number of studies that have applied standardised conditions and adapted, for example, to DNA detection by DAPI labelling or polyP detection by epifluorescence microscopy. The problem that we highlighted at the outset of this work is that the basic conditions (buffer, incubation time, etc.) need to be tested because they are not necessarily transposable from *ad hoc* conditions adapted to microscopy. This is why we took care to test these variables. This work will enable the community to avoid pitfalls (for example, McIlvaine, TE or HEPES buffers are not suitable for the detection of polyP in flow cytometry in combination with DAPI staining). A number of other parameters, such as concentration or storage, had to be tested. Although trivial, they are essential for defining future

45 standardised protocols and must be acquired prior to the studies. The work we have done will save the whole community time and effort in carrying out time-consuming analyses.

This is indicated in the revised manuscript: page 13-lines 448-452.
* * *
**GC3- Reviewer comment** : « *More severely, most of the resulting data is either not shown or not conclusive (see comments*
50 *Figure 2 and 3).* »

**Author's response :** We answered this question at the beginning of our responses to reviewer 1. We agree with this comment and will provide the data and graphs.

All data are presented in the supplemental material of the revised manuscript
* * *
55 **GC4- Reviewer comment** : « *Following their tests, the authors applied their FACS approach to environmental samples and find that the FACS detection highly overestimates the cell numbers/proportions as compared to microscopy (which is the benchmarking tool).* »

**Author's response :** We do not fully understand the meaning of this comment. Throughout the article, we demonstrate that labelling polyP with DAPI is effective for homogeneous samples. On the other hand, we
60 demonstrated the overestimation of polyP after labelling with DAPI in complex environmental samples, which justifies the use of a specific fluorochrome (JC-D7) when dealing with this type of sample. DAPI is a non-specific polyP dye and, in addition to labelling DNA, it also interacts with lipids, displaying metachromatic properties similar to those of polyP (Serafim *et al*., 2002). Although the DAPI-lipid fluorescence is short-lived, with respect to the speed of flow cytometry analysis, it cannot be ignored (P.12 lines 475-476) whereas in microscopy the
65 longer exposure times for counting make it possible to avoid the artefactual counting of lipids.

We indicated in the revised manuscript that "*Controls on the number of polyP cells counted by FCM were carried out by epifluorescence microscopy, a standard method for quantifying and visualizing PAB (Serafim et al., 2002). In almost all the data obtained, the observations by FCM and epifluorescence are significantly different, and*
70 *especially in natural samples, the differences were striking.*" Page 14- Lines 488-491
* * *
**GC5- Reviewer comment** : « *Surprisingly, the authors conclude that their approach is robust. I strongly disagree with this conclusion.* »

**Author's response :** We answered this question at the beginning of our responses to reviewer 1. We agree with
75 this comment and will moderate our assertions in the revised manuscript.

We removed this assumption and we presented a section outlining the pitfalls of the FCM-DAPI labelling approach for polyp. Page 14-Section 4.2.
* * *
**GC6- Reviewer comment** : « *Also, I fail to understand how the authors classified JC-D7 as a useful PolyP-stain when it*
80 *failed to show similar numbers to DAPI (which is also has a questionable PolyP-specificity).* »

**Author's response :** Given its non-specific nature, DAPI leads to an overestimation of polyP detection in flow cytometry (see response to comment GC4). This can be established by comparison with the labelling of polyP with DAPI in epifluorescence microscopy, which was used as a validation method. On the other hand, the counts made with JC-D7 in flow cytometry were not statistically different from those made by epifluorescence microscopy after
85 labelling with DAPI. This leads us to consider that JC-D7 is a useful PolyP-stain for using in flow cytometry. We probably need to explain this aspect further in the revised version of the manuscript for a better understanding.

We have discussed the differences between FCM and epifluorescence microscopy as well as the non-specificity of DAPI in the revised manuscript. Page 14-Section 4.2.

90    **GC7- Reviewer comment** : « *The methods section is not detailed enough to allow for reproduction of the presented approach and tests.* »

**Author's response :** Could you be more specific about the sections on materials and methods that are not detailed enough ?

95    **GC8- Reviewer comment** : *Please also include more literature including (https://enviromicro-journals.onlinelibrary.wiley.com/doi/10.1046/j.1462-2920.2001.00164.x, https://journals.asm.org/doi/10.1128/aem.02592-12, https://doi.org/10.1038/s41396-019-0399-7)*

**Author's response :** The reference to Liu *et al.* 2001 is relevant to our study as they used DAPI staining procedures combined with FISH to identify directly the polyphosphate accumulating traits of different

100   phylogenetic groups. We will cite this article in the discussion section of the revised manuscript. Regarding the references to the work of Martin and Van Mooy (2013) and Fernando *et al.* (2019), if the editors agree that the manuscript should be longer, we could add, in the introduction section, the following information, referring to the publication of Majed et al. (2012) : « Numerous methodologies to quantify and characterise polyP have been developed, including chemical, biological, molecular and microscopic approaches (Majed *et al.* 2012). Most

105   conventional analytical methods (e.g. electron ionisation mass spectrometry) require extensive sample preparation, pre-treatment and pre-fractionation procedures. Advanced analytical techniques, such as nuclear magnetic resonance, Raman, Raman-FISH (Fernando *et al*. 2019) and X-ray spectromicroscopy require much less pre-treatment and allow polyP to be characterised with high molecular and spatial resolution (< to µm). While the potential of these approaches in environmental and biological research is clear, their use remains limited due to

110   the cost and accessibility of analysis instruments. Photometric approaches offer an interesting alternative to the methods discussed above and, the most relevant to date, are based on the interaction between polyP and the fluorochrome 4', 6-diamidino-2-phenylindole (Martin and Van Mooy 2013). »

These informations are provided in the revised manuscript: page 2-lines 59-73

**Responses to specific comments**

115   **SC1- Reviewer comment** : « *Please do not use abbreviations in the abstract* »

**Author's response :** Abbreviations will be removed in the revised manuscript

This has been corrected in the revised manuscript. Page 1-lines 15-28.

**SC2- Reviewer comment** : *It seems that JC-D7 and DAPI could be applied together as their spectrum is different from*
120   *each other. This would also allow to see how specific both stains are.*

**Author's response :**

The spectra of polyP DAPI and JC-D7 overlap particularly well and the acquisition wavelengths are very close indeed (520 nm and 530 nm respectively). Are you referring to the DAPI-DNA spectrum ? If so, the metachromatic effect must be taken into account. It is therefore not possible to carry out a double labelling analysis, DAPI JC-
125   D7. This will probably be possible in the future using deconvolution analysis by spectral cytometry, but it remains

impossible to date using conventional cytometry. If you think it is necessary, we can add the fluorescence spectra of DAPI-green, DAPI-blue and JC-D7 in the supplemental material.

In the revised manuscript, we have added the spectra of the fluorochromes as supplemental material (Figure S1) as well as the following sentence at the end of the discussion :« *The development of cytometry methods, in particular spectral cytometry, could also allow co-labeling of DAPI and JC-D7. This approach is currently impossible with conventional cytometry because the spectra of polyP DAPI and JC-D7 overlap and the acquisition wavelengths are very close (520 nm and 530 nm respectively).* » Page 15-lines 546-549.
* * *
**SC3- Reviewer comment** *: « L65f: add some results please »*

**Author's response :** We will add the following information (in red) : The assays were performed using *Tetrasphaera elongata,* which represent a large part of the microbial biomass (up to 30–35% of the total biovolume of bacteria, Nguyen *et al.* 2011) in enhanced biological phosphate removal systems for wastewater treatment.

Reference : Nguyen et al. 2011 at https://pubmed.ncbi.nlm.nih.gov/21231938/

The following sentence « up to 30–35% of the total biovolume of bacteria, Nguyen *et al.* 2011» has been added in the revised manuscript. Page 3-line 97
* * *
**SC4- Reviewer comment** *: « L77 : I guess it should be MgSO4 x 7 H2O? »*

**Author's response :** Yes, it's a typo that will be corrected in the revised version of the manuscript.

This is corrected in the revised manuscript. Page 3-lines 99-100
* * *
**SC5- Reviewer comment** *: « L901. 1500 x g does not sound much? How were the authors sure this sample did not contain particulate organic matter or non-microbial particles that could cause autofluorescence in the FACS ? »*

**Author's response :** The protocol we are using including the low-speed centrifugation is a classic method used by the scientific community to 'extract' cells from sediments, see for example the publication by Lavergne *et al.* (2014). Of course, there are still organic particles that can lead to aspecific fluorescent labelling in the wavelengths of polyP- DAPI labelling. This is why we use double labelling with SYTO62 to simultaneously label cellular DNA (page 12, lines 452-456).

Reference : Lavergne et al. (2014)- https://www.sciencedirect.com/science/article/pii/S0167701214001870
* * *
**SC6- Reviewer comment** *: « L94: remove "The" at the beginning of the sentence please »*

**Author's response** : 'The' will be removed in the revised manuscript

This has been corrected in the revised manuscript. Page 4-line 121.
* * *
160 **SC7- Reviewer comment** : *« L97f: please remove the '.' Between µg or mg and ml-1. It should be µm ml-1. This appears throughout the manuscript »*

**Author's response** : It will be corrected throughout the manuscript in the revised version.

This has been corrected throughout the revised manuscript.
* * *
165 **SC8- Reviewer comment :** *« L100: what was the solvent used for DAPI stocks and why were they stored at -20°C? »*

**Author's response**

DAPI was prepared according to the supplier's recommendations (Sigma-Aldrich). Solid DAPI (powder) was redissolved (concentration of stock solution = 1mg/ml, i.e. 2.85 mM) in ultrapure water and stored at -20°C.

This has been indicated in the revised manuscript. Page 4-lines 127-129.

170 ---

**SC9- Reviewer comment :** *« L103 : please choose to use either concentration or molarity throughout the methods section »*

**Author's response** : We will use molarity throughout the revised manuscript.

This has been corrected throughout the revised manuscript.
* * *
175 **SC10- Reviewer comment :** *« L113: what does qsp stand for? »*
**Author's response** : We are sorry, it's a francization. It means « sufficient quantity for ». We will correct this in the revised manuscript by indicating that these quantities are given for 1 litre of buffer.

This has been corrected in the revised manuscript. Page 4-lines 143-152.

180 ---

**SC11- Reviewer comment :** *« L120: µm not µM »*
**Author's response** : It will be corrected in the revised manuscript.

185 This has been corrected in the revised manuscript. Page 4-line 152.
* * *
**SC12- Reviewer comment :** *« L125: Darmstadt»*

**Author's response** : It will be corrected in the revised manuscript

190 This has been corrected in the revised manuscript. Page 5-line 160.
* * *
**SC13- Reviewer comment :** *« L126: how were the cells stored at these different temperatures? This is critical information missing »*

**Author's response** : We don't quite understand the meaning of this question. Do you want to know whether cryoprotective agents have been added ?
* * *
**SC14- Reviewer comment :** « *L131: again, please remove the dot '.' between cell and s-1* »

**Author's response** : It will be removed in the revised manuscript.

This has been corrected in the revised manuscript. Page 5-line 168.
* * *
**SC15- Reviewer comment :** « *L129 and L140: is it correct that two different machines were used for the FACS analysis (counting and sorting)?* »

**Author's response** :That is right We only have one machine for cell sorting, the BD FACSAria™ Fusion SORP cell sorter, which is an extremely efficient cytometer for this purpose. However, the FACSAria preparation is cumbersome to set up. Cell counting was therefore performed on a BD LSR Fortessa™ X-20™, which is designed for this purpose. Both instruments have the same lasers and filters, making the analysis comparable, and internal quality control using fluorescent microbeads is used.

This has been explained in the revised manuscript. Page 5 lines 186-190
* * *
**SC16- Reviewer comment :** *« L150f: it sounds as if the cells were filtered onto a black membrane, DAPI-stained, washed and DAPI-stained again. Is this correct? If so, why were the cells stained twice? »*

**Author's response** : This is the protocol that was applied. Polycarbonate black membrane are typically used for DAPI staining (e.g. https://fcelter.fiu.edu/data/protocols/_assets/bacterial_enumeration_protocol.pdf). Double DAPI labelling was performed at two different concentrations to stain PolyP (10 µg. mL$^{-1}$ final concentration) and DNA (1 µg. mL$^{-1}$ final concentration concentration) with a wash between the 2 steps to remove excess DAPI after the first labelling.
* * *
**SC17- Reviewer comment :** « *L153 : Did the authors not use a mounting medium between filters and coverslips (e.g. Citifluor or Vectashield)?* »

**Author's response** : Citifluor or Vectashield are generally used for observations after fluorescent in situ hybridisation (FISH). Classically, for DAPI staining, non-fluorescent immersion oil is deposited on top of the filter and covered with a coverslip. This is what we did.

(for example, see : https://fcelter.fiu.edu/data/protocols/_assets/bacterial_enumeration_protocol.pdf ).

**SC18- Reviewer comment :** *« L165 and also before: please indicate what solution was used to prepare the formaldehyde fixative. Water, 1 x PBS, or something else? Same applies for all stains (e.g. JC-D7) and solutions used. Please be more diligent in adding information »*

**Author's response** We will add the following information in the material and methods section:

JCD7: 10mM in DMSO stock at -20°C then dilution in HEPES buffer

230    DAPI: Solution in H2O 1milligram ml in water, stored at -20°C then dilution in the chosen buffer

Formaldehyde: 37% commercial solution then diluted directly in sample (stabilized with about 10% methanol, Ref Sigma 8.18708).

These informations have been added in the revised manuscript. Page 4-Lines 120-140 and Page 5-lines 162-163.
* * *
235    **SC19- Reviewer comment :** *« L181: it does not become clear what independent staining refers to and how that helped to establish thresholds. Please elaborate. »*

**Author's response. T**he minimum threshold was set for FSC only. The settings for morphological (FSC and SSC) and fluorescence (DAPI, Syto, JC-D7) parameters were then set on the basis of samples unstained or independently stained by the different fluorochromes (SYTO, DAPI).

240    The word "threshold" in cytometry means the signal acquisition threshold ; for the rest, we defined positivity and negativity limits. Each parameter is therefore tested independently to determine where the positivity and negativity limits are for each.

This has been indicated in the revised manuscript. Page 6 lines 192-199
* * *
245    **SC20- Reviewer comment :** *« L185f: what software was used to perform statistical analyses? »*

**Author's response** : We used Graph Pad Prism v8. We will add this information in the revised manuscript.

This has been indicated in the revised manuscript. Page 8 lines 273-279
* * *
**SC21- Reviewer comment :** *« Figure 1: Please try to use another micrograph for Figure 1B where the cells are magnified*
250    *in a similar way as in Figure 1C ».*

**Author's response :** In the revised manuscript, we will use a micrograph for Figure 1B where the cells are magnified in a similar way as in Figure 1C

This micrograph has been provided in the revised manuscript. Page 7-Figure 1.
* * *
255    **SC22- Reviewer comment :** *« L246: It does not become clear what the authors mean with "population structure". Why would the staining buffer interfere with population structure in the first place? »*

As shown in Figure 2A, the dots in the cytograms display a two-cluster structure for the RX strain. These different clusters correspond to cells with different characteristics that affect their position on the cytogram according to

side scatter (SSC). The number of cells in the P2 population increases with time, so the structure (in terms of the number of sub-populations) of the RX population is affected when the strain is analysed in TE buffer.

The term structure has been removed in the revised manuscript. Page 8-Lines 296-301.
* * *
**SC23- Reviewer comment :** *« L254: I am convinced that in the era of almost unlimited online space there is no need for "data not shown" anymore. I would like to ask the authors to show these data as well. »*

**Author's response** : We answered this question in the « responses to general comments" section.

All data are presented in the supplemental material of the revised manuscript
* * *
**SC24- Reviewer comment :** *« L288: why is this reference showing up in the brackets again? It is clear from the methods how the labelling was performed. »*

**Author's response** : The reference will be removed from this sentence in the revised manuscript.

This reference has been removed in the revised manuscript. Page 9-Line 315.
* * *
**SC25- Reviewer comment :** *« L290: Again, it is pretty unacceptable to not show these results. Please add them to the note or to a supplementary file so the readers can judge your conclusions. »*

**Author's response** : We answered this question in the « responses to general comments" section.

All data are presented in the supplemental material of the revised manuscript
* * *
**SC26- Reviewer comment :** *« L290: How did the authors decide if there was a strong or a weak fluorescent signal. This needs to be clarified in order to judge the results, e.g. the data shown in Figure 2C (comparison of HEPES and PBS). »*

**Author's response** : In epifluorescence microscopy, the fluorescence signal is the visual intensity. In flow cytometry it is the MFI (mean of fluorescence intensity) in arbitrary units. In the revised manuscript, we will explain that the word "strong" means positive marking with regard to the limits defined by the controls.

This has been indicated in the revised manuscript. Page 5-Line 184
* * *
**SC27- Reviewer comment :** *« L294f and Figure 1C: The interpretation of these results is pretty unclear to me. The FCM detection of DAPI-stained PolyP signal in the strain that supposedly did accumulate low values of PolyP in HEPES buffer showed a PolyP signal for 99% of the cells. This is surprising. The authors then state that in PBS, the number of cells with a PolyP signal was around 1%. Given the size of the bar in Figure 1C, this value should be around 5-10%. Please clarify. The authors then state that under PBS and microscopy a "correct" ratio was observed which is why they excluded HEPES buffer from further experimentation. First of all, how do the authors know what the correct value is if their "control" strain accumulated PolyP as well? Also, HEPES buffer without any P should in principle be more suited for PolyP detection than*

*PBS, which introduces P to the cells. In addition, why does the microscopy data and the FCM data not agree more with each other? Microscopy should be the control here so this means that FCM highly overestimates the PolyP-detection? »*

**Author's response** : There is a yet unidentified phenomenon that leads to artefactual labelling of RX cells in HEPES buffer in flow cytometry. The experiments replicated several times systematically gave the same result. This artefactual labelling led to an overestimation of the proportion of polyP+ cells within the RX population. What allows us to conclude that these proportions are largely overestimated are the epifluorescence microscopy observations on the same samples. One possible explanation could be that HEPES leads to erroneous measurements of fluorescence parameters assessed by flow cytometry due, for example, to strong alterations of the bacterial membrane due to the composition of this buffer (e.g. lack of divalent cations, Tomasek *et al.* 2018). This buffer is not optimal for diluting the Gram-negative RX strain (leading to artefactual fluorescence detection) and potentially Gram-negative bacteria in general for the detection and quantification of polyP+ cells. This effect was not identified for the Gram positive *T. elongata* strain suggesting that the artefact observed is due to membrane properties.

We will indicate this in the revised manuscript, and show the data for *T. elongata*.

Reference : Tomasek K, Bergmiller T, Guet CC. Lack of cations in flow cytometry buffers affect fluorescence signals by reducing membrane stability and viability of Escherichia coli strains. J Biotechnol. 2018 Feb 20;268:40-52. doi: 10.1016/j.jbiotec.2018.01.008.

This has been indicated in the revised manuscript. Page 13-Lines 158-469
* * *
**SC28- Reviewer comment :** *« Also I would like to ask the authors to also show the data of Figure 2C with TE cells. What are the proportions here? How did HEPES buffer fare in comparison to PBS? »*

**Author's response** : As indicated in the responses to the general comments, we will provide all the necessary data (which will be deposited in FigShare) and graphics in supplementary data of the revised manuscript. The fluorescence shift of cells labelled with DAPI in HEPES compared to those labelled in PBS buffer was not observed for *T. elongata* cells. See also the response to your comment SC27.

All data are presented in the supplemental material of the revised manuscript
* * *
**SC29- Reviewer comment :** *« L304: The data referred to here is not shown »*

**Author's response** : The data will be provided as a supplemental figure in the revised manuscript

All data are presented in the supplemental material of the revised manuscript
* * *
**SC30- Reviewer comment** *: «* *Figure 3B shows the proportion of PolyP detected in cells RX (low-accumulators) for different fixation treatments (8-14%). These proportions are similar to the values shown in Figure 2C where they were referred to as 1%. Again, the data does not really add up and disagrees with the microscopy data shown in Figure 2C. In addition, I want to see the proportion of TE cells according to these tests as they were also counted (see Figure 3A). Why were they not shown in the first place? »*

**Author's response** : Figure 3C shows the % of polyP+ cells in the RX population after fixation with 2% and 4% formaldehyde. These values are compared with the proportions obtained on 'fresh' samples analysed without fixation. As we carried out the experiments using the same RX culture, it is normal that the data are similar between Figure 2c and 3C for the "fresh" sample.

Concerning the difference between the counts by flow cytometry and epifluorescence microscopy, for the RX strain, which we have clearly identified and which we have plotted in figure 2C indicating significant differences. These methods are different. The major advantage of the flow cytometry was its counting speed. Microscopy, in turn, yielded better visualization of the research material. However, despite the variation and substantial subjectivity related to both microscopy and flow cytometry, both methods revealed the proportions of polyP+ cells in a rather congruent manner.

Concerning the TE strain, the data will be provided as a supplemental figure in the revised manuscript

We presented a section outlining the pitfalls of the FCM-DAPI labelling approach for polyp in the revised manuscript. Page 14-Section 4.2. All data are presented in the supplemental material of the revised manuscript
* * *
**SC31- Reviewer comment** *: « L332: I am highly surprised to see that the storage at -80°C did not affect cell morphology, and thus detectability of PolyP. But again the authors chose to not show these data. »*

**Author's response** : As indicated in the « responses to general comments" section, we will provide the data (which will be deposited in FigShare) and graphics in supplementary data of the revised manuscript.

All data are presented in the supplemental material of the revised manuscript
* * *
**SC32- Reviewer comment** *: « L334: How did the authors prepare the 50/50 mixture relative abundance for these tests? »*

**Author's response** : We counted the number of RX and TE cells from each culture using flow cytometry. Using the cell densities obtained for each, we mixed them in a 50 : 50 ratio in abundance. This information will be added in the material and methods section of the revised manuscript.

This has been indicated in the revised manuscript. Page 11. Lines 379-384.
* * *
**SC33- Reviewer comment** *:* *« L383f: The overestimation was already visible in Figures 2 and 3 which indicates that FCM as established in this study is not very useful for the quantitative detection of PolyP accumulating strains »*

**Author's response** : See response to comment SC30

We presented a section outlining the pitfalls of the FCM-DAPI labelling approach for polyp. Page 14-Section 4.2.
* * *
**SC34- Reviewer comment** *:* *« L424: I strongly disagree with the conclusion that the "present study established the basis of a robust protocol for the detection and enrichment of PAB by flow cytometry". The current study, in my opinion, fails to establish the FACS analysis in a comparable manner to fluorescence microscopy which serves as the benchmark. »*

**Author's response** : See response given at the beginning of these "Responses to reviewer 1"

We presented a section outlining the pitfalls of the FCM-DAPI labelling approach for polyp. Page 14-Section 4.2.
* * *
**Responses to reviewer 2**

We would like to thank reviewer 2 for his/her interest in this work and his/her careful and expert reading of the manuscript. As this reviewer pointed out, the limited knowledge on poly-P cycling in natural environments makes such a protocol would be very useful. The comments will enable us to improve the manuscript.

You will find below the answers posted on EGUSPHERE (in black) as well as the corrections made in the revised manuscript (in red).

**Responses to general comments**

**GC1- Reviewer comment** : « *The experimental set-up; using T. elongata as a positive control is excellent and the use of environmental samples is good and necessary since a high-throughput method that is able to detect poly-P in natural samples could potentially give much more insights into the spatial and temporal dynamics of polyphosphate cycling. However, no real negative control has been used. A strain named "RX" was used as a negative control, even though it was shown to accumulate poly-P. Why not use an actual negative control ? The data that has been collected for the "RX" strain could be an excellent addition to show how the protocol functions for species with minor amounts of poly-P, which might be relevant in natural settings.* »

**Author's response :** PolyP occurs as a ubiquitous biopolymer in representatives of all kingdoms of living organisms and every cell type in nature (*LorenzoOrts et al. 2020, New Phytologist ; Akbari et al. 2021, Microbial Biotechnology*). In all prokaryotic cells, polyPs are found in granules. This storage of polyPs can be exacerbated in certain bacteria, which accumulate these polymers to intracellular concentrations in the millimolar range. PolyP represent up to 20% of cell dry weight (Martin *et al*. 2014). This is the case for *T. elongata*, a bacterium that hyperaccumulates polyP and which we used as a « positive control » in this study. As there is no true negative control, prior to this study we screened our bacterial strain library in the laboratory to identify the strain with the lowest accumulation that we used as a « negative control ». We agree that these terms may be ambiguous. To remove any ambiguity, we propose to use the terms « high accumulator » and « low-accumulator » in the revised manuscript to describe the *T. elongata* and RX strains, respectively.

This is corrected in the revised manuscript: page 1-line 24; page 3-lines 100-103; page 8-lines 290.
* * *
**GC2- Reviewer comment** : « *A lot of the data is not shown. Add this data, either to the main results or the supplement.* »

**Author's response :** In order not to make the manuscript too unwieldy, we have chosen not to present certain figures and data, as they do not provide any additional information to that given in the text. It is true, however, that this decision can be embarrassing for the reader and leave some doubt as to the validity of the results. Therefore, in agreement with reviewer 1, all data cited in the manuscript and not included in the main document will be presented in the form of supplementary figures and/or made available on the Figshare repository.

All data are presented in the supplemental material of the revised manuscript
* * *
**GC3- Reviewer comment** : « *The authors claim that this is a robust protocol whereas the data tells a different story. The observed differences between the proportion of poly-P cells detected by either FCM or Epifluorescence microscopy in sediment samples (Fig 2C, Fig 4BEF, Fig 5C and Fig 6C) shows that this is not a robust protocol and that FCM grossly overestimates the amount of cell accumulating poly-P. Overall, I would conclude that the combination of DAPI staining and FCM would not work and that this method should not be used to detect and quantify polyP. However, since DAPI staining combined with*

*fluorescence microscopy is often used to detect polyP, it is still important to get the knowledge out there to show that it does not work when combining with FCM. The data shown here where the dye FC-D7 was used seems to be more promising but requires more data (i.e., fluorescence microscopy to show that it works in prokaryotes) and a similar validation protocol as presented here for DAPI. »*

**Author's response :** Yes, you are right to a certain extent. We have clearly demonstrated that DAPI coupled with cytometry is not a method that can be used for complex environmental samples. We should have been more explicit about the fact that we demonstrate in a step-by-step approach the benefits and pitfalls of polyP detection in flow cytometry. In any case, we show that by using the protocol developed in this study, it is possible to couple flow cytometry and DAPI labelling for polyP detection in homogeneous samples (for example during experiments under controlled conditions on strains in microcosms or mesocosms),

The JC-D7 fluorochrome looks very promising. The community should take advantage of the initial results of our article to test this fluorochrome on their samples to ensure its validity. In the same way that DAPI has been the subject of study after study, JC-D7 will also have to enter this "cycle of testing" by different teams on different types of samples before we can get a clear idea of its value for detecting polyP in environmental microbial samples.

We are the first to use JC-D7 for polyP detection in microbial samples and the first to use it in flow cytometry. JC-D7 has never been used to quantify polyP in prokaryotes using epifluorescence microscopy. We cannot validate an approach by using an approach that has not been validated. This is why we chose to validate the cytometric counts after labelling polyP with JC-D7 by epifluorescence counts after labelling polyP with DAPI. Epifluorescence microscopy for the detection of polyP after labelling with DAPI is an approach validated by the scientific community (e.g. Majed *et al.* 2012).

These aspects have been discussed in the revised manuscript. Page 15- Lines 538-549
* * *
**Responses to specific comments**

**SC1- Reviewer comment** *: « I think the introduction could elaborate a bit more or the functions and ubiquitousness of polyphosphate and put it in an environmental context. Almost all bacteria have the enzymatic potential for poly-P built-up and breakdown and besides being used as a storage compounds/energy reserve, many functions are known and none of them are mentioned in the introduction here. The following papers are excellent references:*
1. *Brown, M. R. W., and Kornberg, A. (2004). Inorganic polyphosphate in the origin and survival of species. Proc. Natl. Acad. Sci. 101, 16085–16087. doi: 10.1073/ pnas.0406909101*
2. *Kornberg, A. (1995). Inorganic polyphosphate - toward making a forgotten polymer unforgettable. J. Bacteriol. 177, 491–496. doi: 10.1128/jb.177.3.491496.1995*
3. *Kornberg, A., Rao, N. N., and Ault-riché, D. (1999). Inorganic polyphosphate: a molecule of many functions. Annu. Rev. Biochem. 68, 89–125. doi: 10.1146/ annurev.biochem.68.1.89*
4. *Rao, N. N., Gómez-García, M. R., and Kornberg, A. (2009). Inorganic polyphosphate: essential for growth and survival. Annu. Rev. Biochem. 78, 605–647. doi: 10.1146/annurev.biochem.77.083007.093039 »*

**Author's response :** Yes, you're right, this information is missing. We suggest introducing the following sentences into the revised manuscript just before talking about hyperaccumulation (p.2 line 43.):

« Likely a key agent in evolution from prebiotic time (Brown and Kornberg 2004; Lorentzo-Orts *et al.* 2020), the functional roles of polyP in the cells of contemporary organisms are many and varied (Konberg *et al.* 1999). PolyP can serve as a source of energy; as a phosphorylating agent for alcohols, including sugars, nucleosides, and proteins; and as a means of activating the precursors of fatty acids, phospholipids, polypeptides, and nucleic acids (Rao *et al.* 2009). »

This information has been added in the revised manuscript. Page 2-Lines 45-50
* * *
**SC2- Reviewer comment** *: « L75: What are "large amounts" of polyP? Is it a large fraction of the cell volume? And if so, how much? Do most cells accumulate poly-P? And if so, how much? »*

455 **Author's response :** We will add the following information : up to 30–35% of the total biovolume of bacteria (Nguyen *et al.* 2011)

This information has been provided in the revised manuscript. Page 3-line 97.
* * *
460 **SC3- Reviewer comment** *: « L78: What is "a very small amount of intracellular polyP? Only few cells or does polyP only take up a small amount of the cell volume? »*

**Author's response :** It's both. We will add this clarification in the revised manuscript

This has been indicated in the revised manuscript. Page 3-Lines 102-103.
* * *
465 **SC4- Reviewer comment** *: « Figure 1A: Beautiful TEM image, the A is on a weird spot in the figure. I would like to see false quantitative color scales for the EDX analyses if that is possible ».*

**Author's response** : We don't really understand the question. There are no false colour scales. A colour corresponds to an element with no differences in shade within the same colour. The relative quantity of the element is determined by the dot density.
* * *
470 **SC5- Reviewer comment** *: « Figure 1BC: Make the figures bigger. »*

**Author's response** : This will be done in the revised manuscript

This has been done in the revised manuscript. Page 7-Figure 1
* * *
**SC6- Reviewer comment** *: « L247: What is meant by "population structure"? »*

475 **Author's response** : As shown in Figure 2A, the dots in the cytograms display a two-cluster structure for the RX strain. These different clusters correspond to cells with different characteristics that affect their position on the cytogram according to side scatter (SSC). The number of cells in the P2 population increases with time, so the structure (in terms of the number of sub-populations) of the RX population is affected when the strain is analysed in TE buffer. If the term "structure" is ambiguous, we can clarify it by simply indicating that the cells are affected.

480 The term structure has been removed in the revised manuscript. Page 8-Lines 296-301.
* * *
**SC7- Reviewer comment** *: « L254 & 255: Show the data somewhere. »*

**Author's response** : We answered this question in the « responses to general comments" section.

All data are presented in the supplemental material of the revised manuscript

485 ---

**SC8- Reviewer comment** *: « L290: Show the data. »*

**Author's response** : We answered this question in the « responses to general comments" section.

All data are presented in the supplemental material of the revised manuscript
* * *
490    **SC9- Reviewer comment** *: « L292 and Figure 2C: There are significant differences between the proportion of polyP+ cells observed by labeling with DAPI in PBS using the FCM or epifluorescence microscopy. Why is this? Isn't this already a first insight the DAPI in combination with FCM is not an appropriate method to detect polyP accumulating bacteria because there are too many false positives ? »*

**Author's response** : Concerning the difference between the counts by flow cytometry and epifluorescence
495    microscopy, for the RX strain, which we have clearly identified and which we have plotted in figure 2C indicating significant differences. These methods are different. The major advantage of the flow cytometry was its counting speed. Microscopy, in turn, yielded better visualization of the research material. However, despite the variation and substantial subjectivity related to both microscopy and flow cytometry, both methods revealed the proportions of polyP+ cells in a rather congruent manner.

500

We presented a section outlining the pitfalls of the FCM-DAPI labelling approach for polyp in the revised manuscript. Page 14-Section 4.2.
* * *
505    **SC10- Reviewer comment** *: « Also, do you have any idea why the use of HEPES would give you so many polyP+ cells? Please elaborate. »*

**Author's response** : There is a yet unidentified phenomenon that leads to artefactual labelling of RX cells in HEPES buffer. The experiments replicated several times systematically gave the same result. This artefactual labelling led to an overestimation of the proportion of polyP+ cells within the RX population. What allows us to
510    conclude that these proportions are largely overestimated are the epifluorescence microscopy observations on the same samples. One possible explanation could be that HEPES leads to erroneous measurements of fluorescence parameters assessed by flow cytometry due, for example, to strong alterations of the bacterial membrane due to the composition of this buffer (e.g. lack of divalent cations, Tomasek *et al.* 2018). This buffer is not optimal for diluting the Gram-negative RX strain (leading to artefactual fluorescence detection) and potentially Gram-negative
515    bacteria in general for the detection and quantification of polyP+ cells. This effect was not identified for the Gram positive T. elongata strain suggesting that the artefact observed is due to membrane properties.

Reference : Tomasek K, Bergmiller T, Guet CC. Lack of cations in flow cytometry buffers affect fluorescence signals by reducing membrane stability and viability of Escherichia coli strains. J Biotechnol. 2018 Feb 20;268:40-52. doi: 10.1016/j.jbiotec.2018.01.008.

This has been indicated in the revised manuscript. Page 13-Lines 158-469
* * *
520

**SC11- Reviewer comment** *: « Figure 3: Both staining periods, DAPI concentrations, and fixation with formaldehyde seem to significantly affect the detection of polyP+ cells with FCM. How was this with epifluorescence microscopy? Is the detection of polyP+ cells using fluorescence microscopy also affected by the staining period and DAPI concentration? This seems especially relevant since the differences in detection by epifluorescence microscopy when compared to FCM all seem
525    to be significant according to the data presented in figure 1C, and later in Figure 4B, E, F, Figure 5B and Figure 6C. Also, even though there are no significant differences observed between fixed and unfixed polyP+ cells for T. elongata (L307), still show the data. »*

**Author's response** : We did not perform detection of polyP+ cells after different DAPI concentrations using epifluorescence microscopy because the aim of this study was to establish conditions for cytometry (including a

530 comparison between different conditions for the FCM approach). Epifluorescence microscopy was the validation method.

For detection differences, see response to comment SC9.

For *T. elongata* data, as indicated in the responses to general comments, all data will be provided in the revised manuscript.

535 ─────────────────────────────────────────────────────

**SC12- Reviewer comment** *: « L332: Show the data. Especially because storage is highly relevant for natural samples. »*

**Author's response** : We answered this question in the « responses to general comments" section.

All data are presented in the supplemental material of the revised manuscript. We have also provided a more explicit figure for the effect of fixation, temperatures and storage time for strain RX. Page 10-Figure 3.

540 ─────────────────────────────────────────────────────

**SC13- Reviewer comment** *: « Figure 4B, 4E, 4F: Where are the statistics between FCM and Epifluorescence? By eyeballing it, the differences seem significant and FCM appears to overestimate the amount of polyP+ cells before sorting and underestimates the amount op polyP+ cells after sorting. »*

**Author's response** : We are unable to produce statistics on these data because the sorting was carried out (as is usually done during cell sorting) on a single sample (50:50 mixture) of RX and *T. elongata* strains.

This has been indicated in the legend of Figure 4 – (Page 11-Line 403) of the revised manuscript

─────────────────────────────────────────────────────

**SC14- Reviewer comment** *: « L337*: What is significant here? Which values are compared? FCM before sorting to FCM after sorting? »

550 **Author's response** : We compared the proportion of polyP+ cells counted by epifluorescence microscopy before and after cell sorting. It is true that the way in which this is presented is not very explicit. This will be corrected in the revised version of the manuscript.

This has been detailed in the revised manuscript. Page 12-Lines 405-411.

─────────────────────────────────────────────────────

555 **SC15- Reviewer comment** *: « L339:* PAB represented less than 10% of the polyP- fraction according to Epifluorescence but the proportion found by FCM is much lower. Why is this? »

**Author's response** :see response to specific comment SC9

We presented a section outlining the pitfalls of the FCM-DAPI labelling approach for polyp in the revised manuscript. Page 14-Section 4.2.

560 ─────────────────────────────────────────────────────

**SC16- Reviewer comment** *: « L378: I do not think that the method has been validated. There are significant differences observed between the FCM and Epifluorescence microscopy in almost every figure shown. The only thing that is validated is that this protocol can be used to enrich polyP+ cells (Figure 4 and Figure 5A). »*

**Author's response** :see response to general comment GC3

565   We presented a section outlining the pitfalls of the FCM-DAPI labelling approach for polyp in the revised manuscript. Page 14-Section 4.2.
* * *
**SC17- Reviewer comment** : *« Figure 5B: This really shows that this is not a robust method to be used for environmental*
570   *samples. »*

**Author's response** : See response to specific comment SC16

We presented a section outlining the pitfalls of the FCM-DAPI labelling approach for polyp in the revised manuscript. Page 14-Section 4.2.
* * *
575   **SC18-  Reviewer  comment** : *« L399-400: Show the data. Was this fluorescence intensity measured with FCM or epifluorescence microscopy. »*

**Author's  response** : Fluorescence intensity was measured with the FCM. The data will be presented in the supplementary material of the revised manuscript

The overlays of cytograms obtained for *T. elongata* after labeling polyP with JC-D7 or DAPI.have been presented
580   in the Figure S.3 of the supplemental material in the revised manuscript.
* * *
**SC19- Reviewer comment** : « **Figure 6**: Why was there no (epi)fluorescence microsocopy performed with JC-D7 labeling? And what is the significance between JC-D7 and DAPI-epifluo in the sediment? Again, the enormous difference between DAPI stained poly+ cells observed via FCM and Epifluorescence show that this is not a robust method. »

585   **Author's response** : see responses to general comments and to specific comment SC16

We have discussed these aspects in the revised manuscript. Page 15-Lines 525-549.
* * *
**SC20- Reviewer comment** : *« The focus of the paper lies on the use of DAPI but looking at Figure 6, the dye JC-D7 looks much more promising for the use in natural samples (or at least for sediment samples). I think validating JC-D7 as a polyP*
590   *specific dye in combination with epifluorescence microscopy that can also be used in prokaryotes would be a big step forward. »*

**Author's response** : We answered this question in the « responses to general comments" section.

We have discussed these aspects in the revised manuscript. Page 15-Lines 525-549.
* * *
**SC21- Reviewer comment** *: « L423: This line does not seem to be relevant here. »*

595   **Author's response** : The sentence will be removed in the revised manuscript.

This sentence has been removed in the revised manuscript. Page 13-Line 444
* * *
**SC22- Reviewer comment** *: « L458-462: I think this is an overestimation of what this protocol can do. In almost all the data shown, the observations with FCM and epifluorescence are significant, and especially in the natural samples, the differences are striking (Figure 5B and Figure 6C). The only thing that is shown is that polyP+ cells can be enriched. I would not recommend DAPI staining in combination with FCM to quantify the amount of polyP accumulating bacteria in a natural sample based on this data. »*

**Author's response** : see response to the reviewer comment GC3

We presented a section outlining the pitfalls of the FCM-DAPI labelling approach for polyp in the revised manuscript. Page 14-Section 4.2.
* * *
**SC23- Reviewer comment** : « **L469-472**: This conclusion cannot be made without comparing the different staining periods and DAPI concentrations with *epifluorescence microscopy.* »

**Author's response** : This conclusion will be removed in the revised manuscript.

Your comment is very pertinent. Given the absence of epifluorescence microscopy controls on the number of PolyP cells as a function of labelling concentration and staining time, we have removed these results from the revised version of the manuscript.
* * *
**SC24- Reviewer comment** : « *L481: Show the results. The publication of negative results is important so other researchers do not have to try it for themselves. »*

**Author's response** : There was really nothing conclusive about these experiments. So there's nothing to show for it.
* * *
**SC24- Reviewer comment** : « *L484-485: The data obtained shows promising results but it does not show the specific nature of PAB labeling. This would require fluorescence microscopy to visualize the JC-D7 dye in combination with for instance SEM-EDS so co-localization analysis can be performed »*

**Author's response** : This would be highly relevant, but it needs to be the subject of research work in its own right, given the specific equipment and the huge amount of development required.

We have discussed these aspects in the revised manuscript. Page 15-Lines 525-549.

---

## Referee Report (RR1)

Review

Technical not: flow cytometry assays for the detection, counting and cell-sorting of polyphosphate-accumulating bacteria

I think the paper in its current state reads very well and is a valuable addition to the literature out there and the information can be used by any researcher interested in polyP counting. It gives all the information required to make a protocol to combine polyP labeling with DAPI and FCM to enrich and isolate high accumulating PAB and gives insights into the use of the JC-D7 dye.

I would therefore recommend it for publication and have only minor comments/suggestions. Another review round is not necessary in my opinion.

Figure 7: Change "Bioavailable inorganique P" in the legend to "Bioavailable inorganic P"
L88: "fluorescence signal" instead of "fluorescent signal"
L271: Change "The counting of DAPI-polyP complexes in epifluorescence microscopy will then be used to validate the cytometric data." to "The counting of DAPI-polyP complexes in epifluorescence microscopy **was** used to validate the cytometric data."
L449: Period after "(Table S.19)"
L509: **Show the data.** There is no limit on the supplement and there are plenty of repositories for microscopy images.
L511: "poly**P"** in the title
L557-558: "causes events with size and structure mimic nonviable cells." should probably be "causes events **which** size and structure mimic nonviable cells." If not, I do not understand this sentence.
L639: "dramatically lower abundance of polyP when comparing to microorganisms" to "dramatically lower abundance of polyP *compared* to microorganisms"

---

## Author Response (AR2)

**RESPONSES TO REVIEWERS**

We thank the reviewers for their careful work. Their comments were constructive and useful, allowing us to incorporate their suggestions into the revised version of the manuscript.
* * *
**Responses to Reviewer 1 general comments**

In his/her general comments, reviewer 1 asked us to :

- **(1)** Provide the taxonomy of the RX strain
- **(2)** Test polyP staining on strains other than RX and *T. elongata*
- **(3)** Extend the polyP labelling tests with JC-D7.

In the revised manuscript, we have addressed all these requests by :

- **(1)** Provide the taxonomy of the RX strain that belongs to the genus *Pseudomonas* (best Blastn match with *Pseudomonas trivialis*) (lines 106-107 of the revised manuscript).

- **(2)** Test of polyP staining with DAPI and JC-D7 on Gram-negative strains belonging to *Acinetobacter lwoffii, Flavobacterium sp*., *Pseudomonas sp.*, *Stenotrophomonas rhizophila* and the Gram-positive strain *Microbacterium hydrocarbonoxydans* DSM 16089 (lines 109-114 and Figure 6 of the revised manuscript, Table S.24 of the supplemental material).

- **(3)** Extension of polyP-JC-D7 labelling assays to include Gram-negative strains belonging to *Acinetobacter lwoffii, Flavobacterium sp*., *Pseudomonas sp.*, *Stenotrophomonas rhizophila* and the Gram-positive strain *Microbacterium hydrocarbonoxydans* DSM 16089 (lines 109-114 and Figure 6 of the revised manuscript, Table S.24 of the supplemental material) but also to include six soil samples (lines 129 to 134, Figure 7 of the revised manuscript ; Table S.25 of the supplementary material). In addition, since the publication of our preprint in Biogeosciences Discussions on 2 May 2024, two articles have been published reporting the use of JC-D7 for the quantification of polyP in microbial samples and also showing that JC-D7 is very promising and specific. We have included this information in the discussion of the revised version of the manuscript (lines 666 to 679 of the revised manuscript).

**Responses to Reviewer 1 specific questions**

**Question 1 :** *« Why was JC-D7 diluted in HEPES buffer when itw as shown to cause problems with strain RX » ?*

**Response:** PolyP labelling with JC-D7 was performed in HEPES buffer as recommended by Angelova *et al.* 2014. We tried to perform polyP-JC-D7 labelling in PBS buffer, but the latter is not optimal for this fluorochrome (lines 457 to 459 of the revised manuscript and Supplementary Figure S.4). In addition, no artefact signals were observed for JC-D7-HEPES labelling for different Gram-negative and Gram-postive strains (Figure 6 of the revised manuscript).
* * *
**Question 2 :** *« L204: I still do not understand why there was the need to stain cells twice. First cells were stained with a high concentration (different conc in the manuscript as given in the response to the reviewers), washed and then stained with a low concentration. The authors argue that the second staining was done to stain the DNA of the cells but that was already achieved during the high concentration staining in my opinion. DAPI does not bind specific to a target based on the concentration. It does not create a problem for the analysis, but this could be unnecessary and introduce bias. »*

**Response :** You are correct, double labelling with DAPI was performed at two different concentrations to stain PolyP (10 µg. mL-1 final concentration) and DNA (1 µg. mL-1 final concentration) with a wash between the two

steps to remove excess DAPI after the first labelling. However, we acknowledge that a single label would have been sufficient.
* * *
**Question 3 :** « *Figure 2: where could the artifact in the McIlvaine buffer come from? The buffer only contains NaH2PO4, citric acid and water. Would it be possible to show the exact same panels for Tris-EDTA for PBS, HEPES, McIlvaine in the supplements? This way one could judge better* »

**Response :** We do not know the origin of the McIlvaine/DAPI artefact reported in the manuscript (lines 558 to 559 of the revised manuscript). The panels for trisEDTA, PBS and HEPES would not be of interest since no events are detected on the cytograms when these buffers are analysed, without cells, with DAPI or with DAPI/SYTO®62.
* * *
**Question 4** : « *Figure 3 and other data show that there cannot be a unified protocol for all bacteria when there is such a high difference between a high accumulator and low accumulator strain ?* »

**Response :** In microbial ecology, we must accept compromises. Microbial communities in natural environments are heterogeneous in terms of physiology, membrane structure, and so on, so it is not possible for one protocol to be optimal for all.

With regard to Figure 3 in particular, the effect of fixation on cytometric analysis of bacteria is well known in ecology (e.g. Kamiya et al., 2007; Troussellier et al., 1995) and a difference in the effect of the fixative between Gram-positive and Gram-negative bacteria has already been observed (Liu et al. 2012). We add this information in the revised manuscript (lines 570 to 572 of the revised manuscript).

Reference : Kamiya, E., Izumiyama, S., Nishimura, M. et al. Effects of fixation and storage on flow cytometric analysis of marine bacteria. J Oceanogr 63, 101–112 (2007).
Troussellier, M., Courties, C. & Vaquer, A. Recent applications of flow cytometry in aquatic microbial ecology. Biology of the Cell 78, 111–121 (1993).
Liu BY, Zhang GM, Li XL, Chen H. Effect of glutaraldehyde fixation on bacterial cells observed by atomic force microscopy. Scanning. 34,6-11 (2012).
* * *
**Question 5** : « *Figure 4: It is shown in Figures 3 and 5 that almost 100% of the T. Elongata cells are repeatedly counted/detected via FCM and epifluorescence microscopy. For this strain a good congruence between FCM and microscopy extist. After mixing the strains T.elongata and RX in equal cell number (50:50 abundance ratio) both methods only detect 36.5 or 12.6 % of PolyP+ cells when it should be approximately 50%. How do the authors explain this discrepancy? Either the cells were not mixed in equal abundance, or the counting does not work anymore as soon as a second bacterial strain is in the mixture which makes the approach not robust enough for any type of mixed communities.* »

**Response :** In some of our experiments, *T. elongata* had almost 100% polyP+ cells. However, variation in the polyP content of microorganisms is a common feature of PAB (e.g. 93.7 ± 1.5 % and 17.5 ± 2.4 % of polyp+ cells in the culture *of T. elongata*; Fig. 5 and Fig. 6 of the revised manuscript), and in the experiment you mentioned, *T. elongata* did not have less than 100 % polyP+ cells ; therefore a ratio < 50 % was normal. We have indicated in the revised version of the manuscript that the proportion of polyp+ can vary for the same strain (lines 651-653 of the revised manuscript).
* * *
**Question 6** : « *Fig. 4: Why was only one sample counted with FCM? This makes it very difficult to judge the result, especially if the goal is to show that FCM is a robust way to count cells after sorting.* »

**Response :** We sorted one sample and then counted it in the FCM. It did not seem appropriate to do this three times.
* * *
**Question 7** : *« Fig5: Here JC-D7 looks promising in the sediment, that would actually be something to extend on »*

**Response :** See the response to the general comments
* * *
**Responses to Reviewer 2 General Comments**

**Comment :** *« L61: Change to: (< µm) »*

**Response :** This point has been changed in the revised manuscript (line 66 of the revised manuscript).
* * *
**Comments :** L86: "*carried out out tests on microbial cells extracted from water and lake sediment samples*" and

L90: « *It also points out that.. »*

**Response :** These points have been changed in the revised manuscript (lines 95-99 of the revised manuscript).
* * *
**Comment :** *« L100: There is no natural negative control. However, it is possible to modify species to get a true negative control by knocking out the ppk gene »*

**Response** : The term « negative control » has been removed in the revised manuscript.
* * *
**Comment : «** *L345 and Figure 2C, L 344-347, L463-465: While I agree with the authors that the HEPES buffer should not be used to analyze DAPI-stained cells with FCM, I would also argue that PBS is not great, at least for the low polyP accumulators and caution should be applied. I do not think that the proportion is in "a simlar range" (L344-L347) (7.2% is very different from 0.9% and accounts for a significant fraction of false positives) and the significant difference observed already shows that this method is problematic for low poly-P accumulating organisms. I would state that the PBS buffer is a better option that the HEPES buffer and therefore, one should not use HEPES but using PBS buffer is already a compromise and also results in artefactual labeling and false positives. »*

**Response** :  This point has been changed in the revised manuscript according to your suggestion : « However, we acknowledge that PBS buffer is already a compromise and also results in artefact labelling and false positives for the 'low polyP accumulation' control strain." (lines 384-385 and lines 565-566 of the revised manuscript).
* * *
**Comment : «** *Figure 3A and 3C: I think it is good that the authors tested the different conditions of fixation, storage time and storage temperature for the RX strain. It is clear that this affects the propotion of polyP cells measured but it is very hard to give any advice on sample preparation if there is no "ground truth" present, i.e, if we do not know the amount of actual poly-P+ cells present (i.e., measured by epifluorescence microscopy). Is it in a similar range as the amount measured in Figure 1C (~0.9% poly-P+ cells)? If so, why does the proportion of polyP+ differ so much from the actual amount? And why does the proportion significantly decrease after fixation with PFA? »*

**Response :** In some of our experiments, *T. elongata* had almost 100% polyP+ cells. However, variation in the polyP content of microorganisms is a common feature of PAB (e.g. 93.7 ± 1.5 % and 17.5 ± 2.4 % of polyp+ cells in *T. elongata* culture ; Fig. 5 and Fig. 6 of the revised manuscript), and in the experiment you mentioned, *T. elongata* did not have less than 100 % polyP+ cells, therefore a ratio < 50 % was normal. We have indicated in the revised version of the manuscript that the proportion of polyP+ can vary for the same strain (lines 651-653 of the revised manuscript).

The effect of fixation on cytometric analysis of bacteria is well known in ecology (e.g. Kamiya et al., 2007; Troussellier et al., 1995) and a difference in the effect of the fixative between Gram-positive and Gram-negative bacteria has already been observed (Liu et al. 2012). We add this information in the revised manuscript ( lines 570 to 572 of the revised manuscript).

Reference : Kamiya, E., Izumiyama, S., Nishimura, M. et al. Effects of fixation and storage on flow cytometric analysis of marine bacteria. J Oceanogr 63, 101–112 (2007).
Troussellier, M., Courties, C. & Vaquer, A. Recent applications of flow cytometry in aquatic microbial ecology. Biology of the Cell 78, 111–121 (1993).
Liu BY, Zhang GM, Li XL, Chen H. Effect of glutaraldehyde fixation on bacterial cells observed by atomic force microscopy. Scanning. 34,6-11 (2012).
* * *
**Comment :** « *Furthermore, RX is a gram-negative strain and it is unclear whether the differences between the treatments are the result of the low polyP accumulation or from the difference in the membrane structure. Overall, it is not possible to draw conclusions about sample preservation on gram negative strains or low polyP accumulators from the data presented here since the testing was done on a low polyP gram-negative strain. To untangle the results, one would have to test the same protocols on a gram-negative strain that accumulates high amounts of poly-P and a gram-positive strain that accumulates very little poly-P.* »

**Response** :In the revised manuscript, we have tested polyP staining with DAPI and JC-D7 on the Gram-negative strains belonging to *Acinetobacter lwoffii, Flavobacterium sp., Pseudomonas sp., Stenotrophomonas rhizophila* and the Gram-positive strain *Microbacterium hydrocarbonoxydans* DSM 16089 ( lines 109-114 and Figure 6 of the revised manuscript, Table S.24 of the supplemental material ).
* * *
**Comment :** « *Line 372: There is no figure 3D, it is only figure 3C.* »

**Response** : This is corrected in the revised manuscript.
* * *
**Comment :** « *L475-480: I agree with what the authors state here but specify that this only works for gram-positive high polyP+ accumulating organisms, it does not work for gram-negative low polyP accumulators and that further testing is needed for gram-positive low polyP accumulators and gram-negative high polyP accumulators.* »

**Response** :We do not understand the question because we have indicated in the manuscript that fixation induces a bias for the Gram-negative strain, but that this is not enhanced by storage at 4°C (lines 569-572 of the revised manuscript).
* * *
**Comments :** « *L475-480: I agree with what the authors state here but specify that this only works for gram-positive high polyP+ accumulating organisms, it does not work for gram-negative low polyP accumulators and that further testing is needed for gram-positive low polyP accumulators and gram-negative high polyP accumulators.* » and « *L499-501: This would only work if the polyP accumulators accumulate high amounts of polyP* »

**Response** :We added the following statement in the revised manuscript : « *Despite the substantial variation and subjectivity associated with microscopy and flow cytometry, both methods revealed, with appropriate labelling (e.g. appropriate staining buffer), fairly congruent proportions of polyP+ cells for high polyP accumulators in homogeneous samples, i.e. strain culture samples.* » (lines 611-613 of the revised manuscript) and « *In light of our data, we acknowledge that this approach will be more effective in the enrichment and isolation of PAB that accumulate large amounts of polyP.* » (lines 624-626 of the revised manuscript).
* * *
**Comment : «** *L510-515: It disagree with the statement that it shows the proportions of polyP+ cells in a congruent manner. To me, it did not show this in a congruent manner for flow cytometry in the RX strain. The differences are significant and a lot of false positives would be measured overestimating the amount of polyP+ cells present. More research would be needed to establish this. For now, the only application is 'simple' matrices in the case of gram-positive high polyP accumulators.* »

**Response** : We have modified this statement in the revised manuscript to read as follows: « *Despite the substantial variation and subjectivity associated with microscopy and flow cytometry, both methods revealed, with appropriate labelling (e.g. appropriate staining buffer), showed the proportions of polyP+ cells to be fairly congruent in the case of high polyP accumulators in homogeneous samples, i.e. strain culture samples. The method combining FCM and DAPI labelling of polyP can therefore be applied to such 'simple' matrices, for example to screen a heterogeneous pool of cells or a single strain for gene-phenotype linkage analyses. The use of flow cytometry combined with DAPI labelling of polyP to count PAB in complex matrices such as sediment or freshwater samples can be used for comparative purposes, but cannot be recommended as a sole method due to high risk of overestimation.* » (lines 611-618 of the revised manuscript).
* * *
**Comment :** L516: « *remove the comma and the however, that way the sentence is much clearer.* »

**Response** : This is corrected in the revised manuscript.
* * *
**Comment : «** *L519: remove the very* »

**Response** : This is corrected in the revised manuscript.
* * *
**Comment :** *L520-521: I would specify that you can effectively enrich and isolate PAB that accumulate large amount of polyP, as this method would not work for low polyP accumulators.*

**Response** : This is indicated in the revised manuscript: "*In light of our data, we acknowledge that this approach will be more effective in the enrichment and isolation of PAB that accumulate large amounts of polyP* ». (lines 624-626 of the revised manuscript)

---

## Author Response (AR3)

**RESPONSES TO REVIEWER 1**

Comment 1 : Abstract: I find the first sentence very confusing. Why would the presence of P-accumulating bacteria invite efforts to reveal "unknown functions" if this manuscript is based on is P-accumulation as the only important function? I believe that the authors can easily rephrase this sentence to make it a bit more meaningful

**Response : This is corrected in the revised manuscript (line 17, page 1).**

Comment 2 : L21: what do the authors mean with multiparametric analysis here?

**Response : You are correct. This is corrected in the revised manuscript (line 20, page 1).**

Comment 3 :L41: I struggle to understand what the "ecological sustainability of P" could mean. The sustainable use of P in agricultural systems? Please clarify?

**Response : The sentence has been reworded to make it clearer:** « *In order to increase the sustainability of P-resources managements, it is crucial to significantly improve our knowledge about the detailed processes, fluxes and reservoirs involved in the geochemical cycle of P. Microorganisms have been shown to be major actors in modern and past geochemical cycles of P either as reservoirs and/or catalysts of processes exchanging P between different reservoirs (Diaz et al. 2008).* » (**lines 41-45, page 2**).

Comment 4 :L70: Should be binding of DAPI to PolyP not vice versa

**Response : This is corrected in the revised manuscript (line 73, page 2).**

Comment 5 :L70-72: Is there a reference that the shift and the intensity is proportional to the amount?

**Response : We have added two references to the revised manuscript (line 75, page 2).**

Tijssen, J. P., Beekes, H. W. and Van Steveninck, J. : Localization of polyphosphates in Saccharomyces fragilis, as revealed by 4',6-diamidino-2-phenylindole fluorescence. Biochim. Biophys. Acta 721, 394–398, doi: 10.1016/0167-4889(82)90094-5, 1982.

Aschar-Sobbi, R., Abramov, A. Y., Diao, C., Kargacin, M. E., Kargacin, G. J., French, R. J. and Pavlov, E. : High sensitivity, quantitative measurements of polyphosphate using a new DAPI-based approach. J. Fluoresc., 18, 859–866, doi: 10.1007/s10895-008-0315-4, 2008.

Comment 6 :L90: In lines 666f it is mentioned that it has recently been used. Please rephrase the sentence in the introduction accordingly

**Response : The sentence has been reworded in the revised manuscript :** 'This novel polyP sensor has been shown to be suitable for staining polyP in living eukaryotic cells and tissues (Angelova *et al.,* 2014), and has recently been used to target polyP in yeast extracts (Deidert *et al.,* 2024) and planktonic environmental samples (Yang *et al.*, 2024).' **(page 3, lines 94-96).**
* * *
Comment 7 :   L92: which can account for up to 30%...

**Response : This is corrected in the revised manuscript (line 97, page 3).**
* * *
Comment 8 :L130: I am not sure what a "parcel" is in this context. Maybe plots?

**Response :We used the term 'parcel' because each parcel corresponds to an area of land with a specific owner and land use (e.g. organic vs. conventional).**
* * *
Comment 9 :L134: incubated may not be the best term here, maybe use "agitated"?

**Response : The sentence has been reworded to make it clearer :** *« To separate the microbial cells from the sediment or soil particles, 10 ml of 0.01 M sodium pyrophosphate buffer (pH 7.2) was added to 1 g of soil or sediment sample in a Falcon® tube and the mixture shaken (280 rpm) at 4 °C for 30 minutes."* **(lines 139-141, page 4).**
* * *
Comment 10 :L130-135: the supernatant probably still contained many clay particles and small POM fragments. Did you verify via microscopy that your sample was consisting of mainly microbes?

**Response No, this has not been done and is not generally done for this type of approach. In any case, there was no dominance of clay particles and small POM fragments in our observations of the sediments using epifluorescence microscopy.**

Comment 11 :L220f: please address the fact that double labelling with DAPI is not necessary somewhere. Either here or in the discussion.

**Reponse : This was noted in the revised manuscript** *« It should be noted that in order to use labelling conditions similar to those used in cytometry (double labelling with polyP/DAPI and SYTO®62/DNA), we used double labelling in epifluorescence microscopy (DAPI/polyP and DAPI/DNA). However, single labelling with a high concentration of DAPI would have been sufficient."* **(lines 228-231 page 6).**
* * *
**RESPONSES TO REVIEWER 2**

Comment 1 : Figure 7: Change "Bioavailable inorganique P" in the legend to "Bioavailable inorganic P".

**Response : This is corrected in the revised manuscript (Figure 7, page 15).**
* * *
Comment 2 : L88: "fluorescence signal" instead of "fluorescent signal"

**Response : This is corrected in the revised manuscript (line 92, page 3).**
* * *
Comment 3 : L271: Change "The counting of DAPI-polyP complexes in epifluorescence microscopy will then be used to validate the cytometric data." to "The counting of DAPI-polyP complexes in epifluorescence microscopy was used to validate the cytometric data."

**Response : This is corrected in the revised manuscript (lines 278-280, page 8).**
* * *
Comment 4: L449: Period after "(Table S.19)"

**Response : This is corrected in the revised manuscript (line 417, page 11).**
* * *
Comment 5 : L509: Show the data. There is no limit on the supplement and there are plenty of repositories for microscopy images.

**Response : We would only have to show photographs of black fields because of the absence of polyP in these crops, which would be of no interest.**
* * *
Comment 6 : L511: "polyP" in the title

**Response : This is corrected in the revised manuscript (line 511, page 14).**
* * *
Comment 7 : L557-558: "causes events with size and structure mimic nonviable cells." should probably be "causes events which size and structure mimic nonviable cells." If not, I do not understand this sentence.

**Response : You are correct. This is corrected in the revised manuscript (line 554, page 15).**
* * *
Comment 8 : L639: "dramatically lower abundance of polyP when comparing to microorganisms" to "dramatically lower abundance of polyP compared to microorganisms"

**Response : This is corrected in the revised manuscript (line 636, page 17).**